

# Damaging viscous-plastic sea ice

Antoine Savard[1] and Bruno Tremblay[1]

[1]Department of Atmospheric and Oceanic Sciences, McGill University, Montréal, QC, Canada

**Correspondence:** Antoine Savard (antoine.savard@mail.mcgill.ca)

**Abstract.** We implement a damage parametrization in the standard viscous-plastic sea ice model to disentangle its effect from model physics (visco-elastic or elasto-brittle vs. visco-plastic) on its ability to reproduce observed scaling laws of deformation. To this end, we compare scaling properties and multifractality of simulated divergence and shear strain rate (as proposed in SIREx1), with those derived from the RADARSAT Geophysical Processor System (RGPS). Results show that including
a damage parametrization in the standard viscous-plastic model increases the spatial, but decreases temporal localization of simulated Linear Kinematic Features, and brings all spatial deformation rate statistics in line with observations from RGPS without the need to increase the mechanical shear strength of sea ice as recently proposed for lower resolution viscous-plastic sea ice models. In fact, including damage an healing timescale of $t_h = 30$ days and an increased mechanical strength unveil multifractal behavior that does not fit the theory. Therefore, a damage parametrization is a powerful tuning knob affecting the
deformation statistics.

## 1   Introduction

It is reasonable to assume that ice could be a material simple enough to describe. After all, it is *just* frozen water. However, this apparent simplicity hides tremendous atomic, chemical, and mechanical complexity. Northern communities succeeded in capturing the spirit of this complexity in their language. The fact that they use numerous rich and precise words for various ilks
of ice and snow reveals a profound implicit understanding of the importance of the symbiotic relation between daily activities and ice identification via both its visual features and its formation (Krupnik, 2010). Ice color, for example, marks the melting zones of sea ice in spring and allows for the identification of hazardous sea ice for walking. Regardless of the beauty and intelligence of this process, other more quantitative metrics are used for problems covering a larger range of scales (from the kilometer scale to thousands of kilometers), including short-term forecast and decadal projections for navigation and global
climate applications.

Sea ice moves under the action of winds and ocean currents, leading to collisions between floes. Internal stresses rapidly redistribute these forces from ice–ice interactions over long distances. Sea ice deformations occur along well-defined lines of deformation called Linear Kinematic Features (LKFs; Kwok, 2001) that are scale-independent and multifractal, ranging from floe size (10 km) to the size of the Arctic Basin, with width ranging from 0 m to 10 km (Hoffman et al., 2019). Along these
lines, sea ice floes can slide along one another (shear), ridge (convergence), or move apart creating leads (divergence). These mechanical processes affect both lead patterns, and the local and pan-Arctic state of the atmosphere-ice-ocean system, notably



the sea ice mass balance, salt fluxes in the upper ocean via brine rejection, and vertical heat and moisture fluxes between the ocean and the atmosphere (Aagaard et al., 1981; McPhee et al., 2005). As such, their multifractality and scaling properties are important to capture in a sea ice model for all applications.

Statistical properties derived from Synthetic Aperture Radar (SAR) imagery of Arctic sea ice show that LKFs exhibit complex laws, including spatiotemporal scaling (e.g. Marsan et al., 2004; Marsan and Weiss, 2010; Rampal et al., 2008). These statistical characteristics are theorized to result from brittle compressive shear faults (Schulson, 2004), and a cascade of fracture that redistributes stresses within the pack ice (e.g. Marsan and Weiss, 2010; Dansereau et al., 2016). The complexity of these interactions is undeniable, and a desirable sea ice model for the Arctic system should represent LKFs adequately.

Dynamical sea ice models use a diverse range of rheologies to simulate sea ice motion. A rheology describes the relationship between internal stress and deformation (rate) for a given material. In the standard viscous-plastic (VP) rheology — elliptical yield curve and normal flow rule (e.g. Hibler, 1979, and its variants) —, sea ice is considered as a highly-viscous fluid for small deformations. In this case, sea ice deforms as a creeping material. When a critical threshold in shear, compression and tension, defined by the yield curve, is reached, the ice fractures and enters a plastic regime (larger, permanent, rate-independent

deformation). The main advantage of using a viscous-plastic model over a more physical elastic-plastic (EP) model (e.g. Coon et al., 1974) is that the material has no "memory" of past deformations and it is not necessary to keep track of all the previous strain state, rendering the VP formulation mathematically and numerically simpler. Since the first formulation of the VP model, much work has been done to improve the efficiency of the numerical solver used to solve the highly non-linear momentum equations (Hunke and Dukowicz, 1997; Hunke, 2001; Lemieux et al., 2008; Lemieux and Tremblay, 2009; Lemieux et al.,

2010; Bouillon et al., 2013).

Following a reassessment of basic (incorrect) assumptions behind models developed from the Arctic Ice Dynamics Joint EXperiment (AIDJEX) (sea ice is isotropic and has no tensile strength, Coon et al., 1974, 2007) new rheologies are proposed to mend some of these problems. For instance, ice would be better described with the inclusion of deformation on discontinuities, and an anisotropic yield curve with tension (Coon et al., 2007). Models that incorporate some of these recommendations include

the Elasto-Brittle and modification thereof (EB, MEB, and BBM: Girard et al., 2011; Dansereau et al., 2016; Olason et al., 2022) Finite Element Models (FEM), in which elastic deformations are followed by brittle failure, while larger deformations along fault lines following damage build-up are viscous. These models include a damage parametrization that accounts for the fact that damage associated with (prior) fractures also affects ice strength in addition to ice thickness and concentration (see, for example, Girard et al., 2011; Rampal et al., 2016; Dansereau et al., 2016; Olason et al., 2022). These authors argued

that the inclusion of a damage parametrization was a key factor for the proper simulation of sea ice deformations that follows observed spatial and temporal scaling properties (see also Dansereau et al., 2016). In other models (e.g. Elastic-Anisotropic-Plastic (EAP), Tsamados et al., 2013; Wilchinsky and Feltham, 2006), the fracture angle between conjugates pairs of LKFs is specified, leading to anisotropy between interacting diamond-shaped floes. Other approaches include the elastic-decohesive rheology using a material-point method (Schreyer et al., 2006; Sulsky and Peterson, 2011), in which the lead mechanics are

simulated through decohesion.





Damage parametrizations — first developed in rock mechanics — are ad-hoc in that they are not derived from observations and/or from first physics principle. For instance, a damage parameter can be quantitatively expressed as a scalar relationship between the elastic modulus of a material before and after fracture (Amitrano et al., 1999). In this model, the ice strength does not decrease when damage is present; instead, it is the Young's modulus that decreases, resulting in larger deformation
for the same stress state. This was put to advantage in the EB model family where the damage is expressed as a function of the (time-step dependant) stress overshoot in principal stress space referenced to a yield criterion (Rampal et al., 2016; Plante et al., 2020). Another approach used in rock mechanics first considers mode I (tensile) failure on the plane where the maximum tensile stress occurs, followed by crack propagation along the plane where the mode II (shear) stress intensity factor is maximized (Isaksson and Ståhle, 2002a, b). Other more complex descriptions of damage in brittle materials such as fracture
initiation around elliptical flaws are used in rock mechanics (e.g. Hoek, 1968) and could in principle be implemented in sea ice models.

Earlier model–observation comparison studies, aimed at defining the most appropriate rheology for sea ice, found that any rheological model that includes compressive and shear strength reproduces observed sea ice drift, thickness, and concentration equally well (e.g. Flato and Hibler, 1992; Kreyscher et al., 2000; Ungermann et al., 2017). The modeling community subse-
quently used deformation statistics (probability density function, spatiotemporal scaling, and multifractality) to discriminate between different sea ice rheological models (Marsan et al., 2004). Results from the community-driven Sea Ice Rheology Experiment (SIREx), under the auspice of the Forum for Arctic Modeling and Observational Synthesis (FAMOS), showed that any model with a sharp transition from low (elastic or viscous creep) deformations to large (plastic or viscous) deformations can reproduce the new deformation-based metrics — provided the models are run at sufficiently high resolution: 2 km for
Finite Difference Models (FDM), and 10 km for FEM (Bouchat et al., 2022). A last unsuccessful attempt at discriminating between rheological models includes the analysis of the LKF density and angles of fracture between conjugate pairs of LKFs; to this point, all rheologies overestimate the angles of fracture and all reproduce densities of LKF comparable to observations provided a small enough resolution is used (2 km for FDM, and 10 km for FEM) (Hutter et al., 2021).

Ultimately the best way to compare models is to isolate one aspect between two different models. An important step toward
this goal was the implementation of the MEB rheology in finite difference, allowing for a direct comparison between VP and MEB rheologies in the same numerical framework (Plante et al., 2020). Other significant differences between the VP and MEB models include the sub-grid-scale damage parametrization and the consideration of elastic deformations prior to fracture allowing the material to retain a memory of past deformations. In an attempt to further disentangle the effect of elasticity, damage and discretization, we include a damage parametrization in the standard VP model, following recommendations from
SIREx (Bouchat et al., 2022), and Olason et al. (2022). To this end, we compare both simulated (with and without damage) and the RADARSAT-derived Eulerian deformation products using probability density functions (PDFs), spatiotemporal scaling laws, and multifractality.

The paper is organized as follows. First, we describe the model in section 2. Then we introduce a damage parametrization that can be used in the context of a viscous plastic model. The sea ice deformation data and deformation metrics used to evaluate



the model's performance are described in sections 3 and 4. Results and discussion of the results are presented in sections 5 and 6. Finally, concluding remarks and directions for future work are summarized in section 7.

## 2 Models

### 2.1 Governing Equations

The two-dimensional equation governing the temporal evolution of sea ice motion is given by:

$$m\left[\frac{\partial \boldsymbol{u}}{\partial t} + (\boldsymbol{u}\cdot\boldsymbol{\nabla})\boldsymbol{u}\right] = -mf\,\hat{\boldsymbol{k}}\times\boldsymbol{u} + \boldsymbol{\tau}_{\mathrm{a}} + \boldsymbol{\tau}_{\mathrm{w}} - mg\boldsymbol{\nabla}H_{\mathrm{d}} + \boldsymbol{\nabla}\cdot\boldsymbol{\sigma}, \tag{1}$$

where $m\,(=\rho_{\mathrm{i}}h)$ is the sea ice mass per unit area, $\rho_i$ is the ice density, $h$ is the mean ice thickness, $\boldsymbol{u}\,(=(u,v))$ is the horizontal ice velocity vector, $\hat{\boldsymbol{k}}$ is a unit vector perpendicular to the sea ice plane, $f$ is the Coriolis parameter, $\boldsymbol{\tau}_{\mathrm{a}}$ is the surface wind stress, $\boldsymbol{\tau}_{\mathrm{w}}$ is the water drag, $g$ is the gravitational acceleration, $H_{\mathrm{d}}$ is the sea surface dynamic height, and $\boldsymbol{\sigma}$ is the vertically integrated internal ice stress tensor. In the following, the advection term is neglected because it is orders of magnitude smaller

than the other terms for a 10-kilometer spatial resolution (Zhang and Hibler, 1997). The surface air stress and water drag are parametrized as quadratic functions of the ice velocities with constant turning angle $(\theta_{\mathrm{a}}, \theta_{\mathrm{w}})$ for the atmosphere and the ocean (e.g. McPhee, 1975, 1986; Brown, 1979):

$$\boldsymbol{\tau}_{\mathrm{a}} = \rho_{\mathrm{a}}C_{\mathrm{a}}|\boldsymbol{u}_{\mathrm{a}}^{\mathrm{g}}|\left(\boldsymbol{u}_{\mathrm{a}}^{\mathrm{g}}\cos\theta_{\mathrm{a}} + \hat{\boldsymbol{k}}\times\boldsymbol{u}_{\mathrm{a}}^{\mathrm{g}}\sin\theta_{\mathrm{a}}\right), \tag{2}$$

$$\boldsymbol{\tau}_{\mathrm{w}} = \rho_{\mathrm{w}}C_{\mathrm{w}}|\boldsymbol{u}_{\mathrm{w}}^{\mathrm{g}} - \boldsymbol{u}|\left[(\boldsymbol{u}_{\mathrm{w}}^{\mathrm{g}} - \boldsymbol{u})\cos\theta_{\mathrm{w}} + \hat{\boldsymbol{k}}\times(\boldsymbol{u}_{\mathrm{w}}^{\mathrm{g}} - \boldsymbol{u})\sin\theta_{\mathrm{w}}\right], \tag{3}$$

where $\rho_{\mathrm{a}}$ and $\rho_{\mathrm{w}}$ are the air and water densities, $\boldsymbol{u}_{\mathrm{a}}^{\mathrm{g}}$ and $\boldsymbol{u}_{\mathrm{w}}^{\mathrm{g}}$ are the geostrophic winds and ocean currents, and $C_{\mathrm{a}}$ and $C_{\mathrm{w}}$ are the air and water drag coefficients. The reader is referred to Tremblay and Mysak (1997) and Lemieux et al. (2008, 2010) for more details on the model and the numerical solver.

The constitutive law for the standard viscous-plastic rheology with an elliptical yield curve and associated (normal) flow rule can be written as, (Hibler, 1977, 1979),

$$\sigma_{ij} = 2\eta\dot{\varepsilon}_{ij} + (\zeta - \eta)\dot{\varepsilon}_{kk}\delta_{ij} - \frac{P_r}{2}\delta_{ij}, \tag{4}$$

where $P_r/2$ is a replacement pressure term and $\zeta$ and $\eta$ are the nonlinear bulk and shear viscosities defined as:

$$\zeta = \frac{P}{2\Delta}, \tag{5}$$

$$\eta = \frac{\zeta}{e^2}, \tag{6}$$

$$\Delta = \left[(\dot{\varepsilon}_{11} + \dot{\varepsilon}_{22})^2 + e^{-2}(\dot{\varepsilon}_{11} - \dot{\varepsilon}_{22})^2 + 4e^{-2}\dot{\varepsilon}_{12}^2\right]^{1/2}. \tag{7}$$

The sea ice pressure $P$ is parametrized as:

$$P = P^* h \exp\left\{-C(1-A)\right\}, \tag{8}$$



where $P^*$ ($= 27.5 \times 10^3 \, \text{N/m}$) is the ice strength parameter, $A$ is the sea ice concentration, and $C$ ($= 20$) is the ice concentration parameter, an empirical constant characterizing the dependence of the compressive strength on sea ice concentration (Hibler, 1979). For small strain rates ($\Delta \longrightarrow 0$), the viscosities tend to infinity, and the bulk and shear viscosities $\zeta$ and $\eta$ are capped to a maximum value using a continuous version of the classical replacement scheme (Hibler, 1979; Lemieux and Tremblay, 2009):

$$\zeta = \zeta_{\text{max}} \tanh \left( \frac{P}{2\Delta \, \zeta_{\text{max}}} \right), \tag{9}$$

where $\zeta_{\text{max}} = 2.5 \times 10^8 P$ (Hibler, 1979), equivalent to a minimum value of $\Delta_{\text{min}} = 2 \times 10^{-9} \, \text{s}^{-1}$ (Kreyscher et al., 1997). In the limit where $\Delta \longrightarrow \infty$ ($x \longrightarrow 0$), $\tanh x \approx x$, and Equation 9 reduces to $\zeta = P/2\Delta$ (Equation 5). In the limit where $\Delta \longrightarrow 0$ ($x \longrightarrow \infty$), $\tanh x \longrightarrow 1$, and $\zeta = \zeta_{\text{max}}$. The replacement pressure $P_r$ is given by

$$P_r = 2\zeta\Delta, \tag{10}$$

which ensures a smooth transition between the viscous and plastic regimes, and stress states that lie on ellipses that all pass through the origin.

## 2.2 Damage Parametrization

### 2.2.1 Background

Progressive damage models were initially developed to model the nonlinear brittle behavior of rocks (Cowie et al., 1993; Tang, 1997; Amitrano and Helmstetter, 2006). Since then, many studies integrated some damage mechanism in which the mechanical ice properties (e.g., elastic stiffness $E$ and viscous relaxation time $\eta$ and $\lambda$) are written in terms of a scalar, non-dimensional parameter $d$ that represents the sub-grid scale damage of the ice (Girard et al., 2011; Dansereau et al., 2016; Rampal et al., 2016; Plante et al., 2020). For example, Dansereau et al. (2016) proposed the following parametrization of the elastic stiffness ($E$) and the viscosity ($\eta$) akin to the ice pressure in Hibler (1979):

$$E = E_0 h \exp\{-C(1-A)\}(1-d(t)), \tag{11}$$

$$\eta = \eta_0 h \exp\{-C(1-A)\}(1-d(t))^\alpha, \tag{12}$$

$$\frac{\eta}{E} = \lambda = \frac{\eta_0}{E_0}(1-d(t))^{\alpha-1}, \tag{13}$$

where $E_0$ and $\eta_0$ are the (constant) Young's modulus and viscosity of undeformed ice, and $\alpha$ ($> 1$) is a parameter that controls the rate at which the viscosity decreases and the ice loses its elastic properties. In this formulation, $E$ and $\eta$ depend on their undamaged value ($E_0$ and $\eta_0$), sea ice thickness and concentration ($A$ and $h$), and a time-dependent damage ($d(t)$).

In progressive damage parametrization, damage builds as a function of the stress overshoot beyond the yield curve. Following Plante and Tremblay (2021), the scaling factor $\Psi$ ($0 < \Psi < 1$) required to bring a super-critical stress ($\sigma'$) state back on the yield curve ($\sigma^f$) is written as:

$$\boldsymbol{\sigma}^f = \Psi \boldsymbol{\sigma}', \tag{14}$$





where $\boldsymbol{\sigma}^f$ is the corrected stress. The corrected state of stress $(\sigma_1^f, \sigma_2^f)$ is defined as the intersection point of the line joining $(\sigma_1', \sigma_2')$ and the failure envelope of the Mohr-Coulomb criterion along any stress correction path. Note that the stress correction path is not a flow rule; it does not change the visco-elastic constitutive equation of the MEB model. Its purpose is to convert the excess stress into damage ($d$). This definition of damage assumes that only stresses change post-fracture, and the strain (rate) does not. The evolution equation for the damage parameter can be written as (Dansereau et al., 2016; Plante et al., 2020):

$$\frac{\mathrm{d}}{\mathrm{d}t}d = \frac{(1-\Psi)(1-d)}{t_\mathrm{d}} - \frac{1}{t_\mathrm{h}}, \tag{15}$$

where $t_\mathrm{d}\,(=\mathcal{O}(1)\,\mathrm{s})$ and $t_\mathrm{h}\,(=\mathcal{O}(10^5)\,\mathrm{s})$ are the damage and healing timescales, and the condition $\Delta t \ll \lambda$ must be met for stability reason (Dansereau et al., 2016). Consequently, the damage at any given time is a function of the previously accumulated damage. This constitutes the memory of the previous stress state in the MEB model.

### 2.2.2 New VP Model Damage Parametrization

In the standard VP model, the ice strength $P$ depends only on the ice concentration $A$ and the ice mean thickness $h$. Sea ice, therefore, weakens only when sea ice divergence is present along an LKF — affecting the ice strength through the exponential dependence on the sea ice concentration (Equation 8) — contrary to real sea ice that weakens when a fracture is present irrespective of whether post fracture divergence or convergence is present.

We include damage in the VP model (akin to what is used in the MEB formulation) using a simple advection equation with source/sink terms of the form:

$$\frac{\partial d}{\partial t} + \boldsymbol{\nabla} \cdot (\boldsymbol{u}d) = \frac{1 - (\zeta/\zeta_\mathrm{max})^{1/n} - d}{t_\mathrm{d}} - \frac{d}{t_\mathrm{h}}, \tag{16}$$

which asymptotes to the steady state solution $d = 1 - (\zeta/\zeta_\mathrm{max})^{1/n}$, — a generalization of the damage parameter for VP models proposed by Plante (2021) — in the absence of advection and healing, and exponentially decays to zero when only healing is considered. In contrast with the MEB model, damage is not bound by the propagation speed of elastic waves. We choose $t_\mathrm{d}$ ($= 1$ day) and $t_\mathrm{h}$ (ranging from 2 to 30 days) as typical times scales for fracture propagation and healing (see Dansereau et al., 2016; Murdza et al., 2022, for small healing timescale explanations). The choice of a small damage timescale comes from the synoptic timescale at which fractures develop, while a large healing timescale comes from the thermodynamic growth of one meter of ice. Note that a VP model is a nearly ideal plastic material, i.e. it can be considered as an elastic-plastic material with an infinite elastic wave speed; therefore, the fracture propagation is instantaneous (i.e., it is resolved with the outer loop solver of an implicit solver or the sub-cycling of an EVP model). In the above equation, $n$ is a free parameter setting the steady-state damage for a given deformation state. Using Equation 9, and the fact that $\zeta_\mathrm{max} = P/2\Delta_\mathrm{min}$, Equation 16 can be written as:

$$\frac{\partial d}{\partial t} + \boldsymbol{\nabla} \cdot (\boldsymbol{u}d) = \frac{1 - \tanh^{1/n}(\Delta_\mathrm{min}/\Delta) - d}{t_\mathrm{d}} - \frac{d}{t_\mathrm{h}}. \tag{17}$$

Following Dansereau et al. (2016); Rampal et al. (2016), the coupling between the ice strength and the damage is written as,

$$P = P^* h \exp\{-C(1-A)\}(1-d), \tag{18}$$



where $P$ varies linearly with $d$, and where $d$ incorporates the full non-linearity of the viscous coefficients ($\zeta$). We refer to this model as VPd in the following.

## 2.3 Forcing, Domain, and Numerical Scheme

The model is forced with 6-hourly geostrophic winds calculated using sea level pressure (SLP) from the National Centers for Environmental Prediction/National Center for Atmospheric Research (NCEP/NCAR) reanalysis (Kalnay et al., 1996). First, SLPs are interpolated at the tracer point on the model C-grid using bicubic interpolation (Akima, 1996). The field is then smoothed using a gaussian filter with $\sigma = 3$, and the geostrophic winds are computed from the smoothed field, yielding winds on the model's B-grid. The winds are interpolated linearly in time to get the wind forcing at each time step. The model is coupled thermodynamically to a slab ocean. The climatological ocean currents were obtained from the steady-state solution of the Navier–Stokes equation with a quadratic drag law, without momentum advection, assuming a two-dimensional, non-divergent velocity field and forced with a 30-year climatological wind stress field. Monthly climatological ocean temperatures are specified at the model's open boundaries from the Polar Science Center Hydrographic Climatology (PHC 3.0) (Steele et al., 2001). The reader is referred to Tremblay and Mysak (1997) for more details.

The equations are solved on a cartesian plane (polar stereographic projection) with a regular 10 km grid. The equations are discretized on an Arakawa C-grid and solved at each time step ($\Delta t = 1$ hour) using the Jacobian Free Newton-Krylov (JFNK) method (Lemieux et al., 2010). At each Newton Loop (NL) of the solver, the linearized set of equations is solved using a line successive over-relaxation (LSOR) preconditioner, and the Generalized Minimum RESidual (GMRES) method (Lemieux et al., 2008) with a relaxation parameter $\omega_{\mathrm{lsor}} = 1.3$. The non-linear shear and bulk viscosity coefficients and the water drag are then updated, and the process is repeated using an inexact Newton's method until either the total residual norm of the solution reaches a user-defined value ($\gamma = 10^{-2}$) or the maximum number of Newton Loop is reached ($\mathrm{NL}_{\mathrm{max}} = 200$) (Lemieux et al., 2010).

Following Bouchat and Tremblay (2017), the model is first spun-up (with damage turned off), with a set of ten random years between 1970 and 1990, a constant one-meter ice thickness, and 100% concentration as initial conditions. The shuffling of the spin-up years is used to prevent biases associated with low-frequency variability, such as the Arctic Oscillations or Arctic Ocean Oscillations (Thompson and Wallace, 1998; Rigor et al., 2002; Proshutinsky and Johnson, 2011). From the spun-up state, each simulation is run from January 1, 2002, to January 31, 2002. The deformations statistics presented below are robust to the exact choice of winter (Bouchat and Tremblay, 2017).

Both the control and simulation with damage use the same initial conditions. In order to test the sensitivity of the results to the choice of initial conditions, the model was spun up for one additional year including the damage parametrization (recall that the healing timescale is 30 days) and the simulations were repeated. The results presented below are also robust to the exact choice of initial conditions.





## 3 Observations

215 We use the three-day gridded sea ice deformation from the Sea Ice Measures dataset, formerly called RADARSAT Geophysical

Processor System (and referred to as RGPS in the following for simplicity) (Kwok et al., 1998; Kwok, 1997). The RGPS data

set is obtained from Lagrangian ice velocity fields by tracking the corners of initially uniform grid cells on consecutive synthetic

aperture radar (SAR) images. The deformation of the grid cells is used to approximate the velocity derivatives and the strain

rate invariants $\varepsilon_I$ and $\varepsilon_{II}$ using line integrals (Kwok et al., 1998). The initial Lagrangian grid spatial resolution is 10 km $\times$ 10

220 km, except in a 100 km band along the coasts, where a coarser resolution of 25 km is used. Finally, the data is regridded onto a

12.5 km $\times$ 12.5 km fixed polar stereographic projection using a three-day temporal resolution. The three-day gridded data set

is available from 1997 to 2008 for summer and winter (November to July) on the ASF DAAC website (https://asf.alaska.edu/).

Following Bouchat and Tremblay (2017), we only use strain rates larger than $|0.005|$ day$^{-1}$ — equal to the tracking error of

about 100 m on the vertices of the Lagrangian grid cells (Lindsay and Stern, 2003).

## 4 Methods

225

Following Bouchat and Tremblay (2017), Hutter et al. (2018), Girard et al. (2009), and Marsan et al. (2004), we compare the

probability density functions, spatiotemporal scaling laws of the mean deformation rates, and multifractal properties simulated

by the model with the RGPS data (see section 4.1 to 4.4 below for details). We calculate all metrics inside the SAR sea ice

RGPS data where an 80% temporal data coverage is present for the winters 1997–2008 — referred to as RGPS80 in the

230 following (see Figure 1 or Bouchat and Tremblay, 2017).



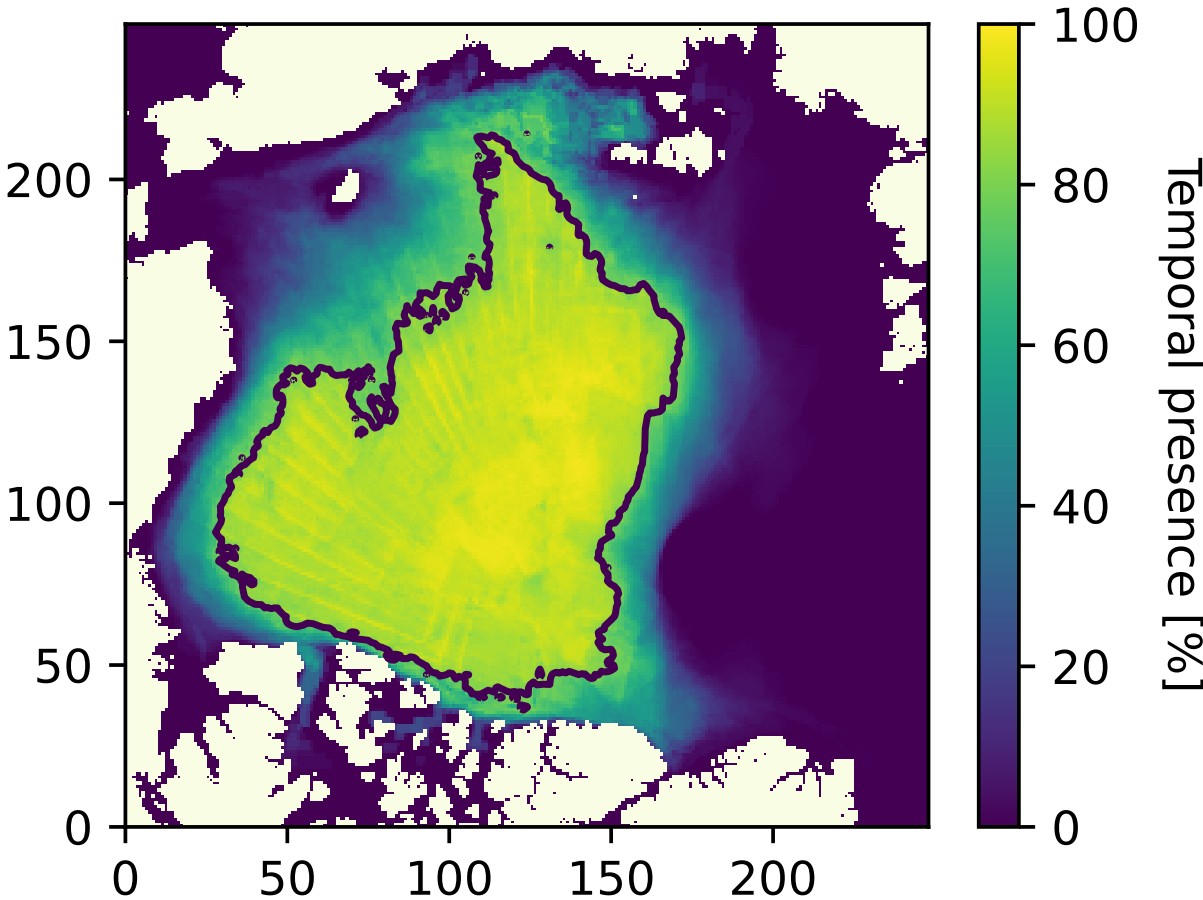

**Figure 1.** Hotness map of temporal presence in the RGPS observations for January 2002. The black line represents the RGPS80 mask and is drawn at the 80% temporal frequency contour. This mask is used for all results.

## 4.1  Simulated Deformation Fields

Following Marsan et al. (2004) and Bouchat and Tremblay (2017), the total sea ice deformation rates are calculated from the (hourly) divergence ($\dot{\varepsilon}_\mathrm{I}$) and the maximum shear strain rate ($\dot{\varepsilon}_\mathrm{II}$) as:

$$\dot{\varepsilon}_\mathrm{total} = \sqrt{\dot{\varepsilon}_\mathrm{I}^2 + \dot{\varepsilon}_\mathrm{II}^2}, \tag{19}$$





where

$$\dot{\varepsilon}_{\mathrm{I}} = \frac{\partial u}{\partial x} + \frac{\partial v}{\partial y}, \tag{20}$$

$$\dot{\varepsilon}_{\mathrm{II}} = \sqrt{\left(\frac{\partial u}{\partial x} - \frac{\partial v}{\partial y}\right)^2 + \left(\frac{\partial u}{\partial y} + \frac{\partial v}{\partial x}\right)^2}. \tag{21}$$

The sea ice velocities are first averaged over a period of three days in order to match the temporal resolution of the RADARSAT observations. The averaged velocity fields are then used to calculate the strain rate invariants at the center of each grid cell. These values represent averaged Eulerian deformation rates over the grid cells area.

### 4.2 Probability Density Functions (PDFs)

Probability density functions are used to assess the ability of the models to reproduce large deformation rates and to determine their statistical distribution. We separate the domain into logarithmically increasing bins and perform a least-square power-law fit on the tail of the log–log distributions where the interval for a given model consists of all bins up to an order of magnitude from the largest deformation bin available. Therefore, intervals between runs differ, but each interval is the most representative of the deformation decay for a given model (Bouchat et al., 2022). To quantify the difference between the shape of the simulated and observed PDFs, we use the Kolmogorov-Smirnov (KS) distance $D$, defined as the absolute difference between the cumulative density functions (CDFs) of the models $C_m(\dot{\varepsilon}_n)$ and the data $C_d(\dot{\varepsilon}_n)$:

$$D = \max_{\dot{\varepsilon}_n \geq \dot{\varepsilon}_{n,\mathrm{min}}} |C_m(\dot{\varepsilon}_n) - C_d(\dot{\varepsilon}_n)|. \tag{22}$$

In this approach, the shape of the PDF is taken into account directly and there is no need to a priori assume the underlying statistical distribution of the PDF. The interpretation of the KS-distance is straightforward: a smaller $D$ implies a closer agreement between observed and simulated statistical distributions.

As noted in Bouchat and Tremblay (2020) and Bouchat et al. (2022), a linear decay in deformations does not imply a power law, as other distributions (e.g., log-normal distributions) can also approximately decay linearly (Clauset et al., 2009). Therefore, we do not assume that the power-law exponents derived from the CDFs are representative of the true distributions; we instead use them as a means to differentiate between simulated and observed PDFs of deformation rates. We therefore use the average of the absolute difference of the logarithms of the simulated and observed PDFs (see also Bouchat et al., 2022). This metric has the advantage of giving more weight to the tail of the PDFs (small probabilities, large deformation rates). Finally, we present results for negative and positive divergence separately to avoid error cancellation (Bouchat et al., 2022).

### 4.3 Spatiotemporal Scaling Analysis

Following Marsan et al. (2004), we use the following coarsening algorithm to compute the spatiotemporal scaling exponent of the mean deformation rates derived from models and RGPS observations to estimate the scaling exponents:

$$\langle \dot{\varepsilon}_{\mathrm{tot}}(L,T)\rangle \sim L^{-\beta(T)}, \tag{23}$$

$$\langle \dot{\varepsilon}_{\mathrm{tot}}(L,T)\rangle \sim T^{-\alpha(L)}, \tag{24}$$





where $L$ and $T$ are the spatial and temporal scales at which sea ice total deformation rates are averaged, and $\beta$ and $\alpha$ are the spatial and temporal scaling exponents. As pointed out by Weiss (2017), $\beta$ can take values between 0 (homogeneous deformations) and 2 (deformations concentrated in a single point), while $\alpha$ can take values between 0 (random deformation events) and 1 (one single extreme event).

We find $\beta$, by first averaging the simulated velocity fields to match the 3-day temporal aggregate of RGPS. We then compute

the mean ice velocities in boxes of varying sizes $L$ from that of the models' spatial resolution (10 km) to the full domain size with doubling steps: $L = 10, 20, 40, 80, 160, 320, 640$ km. The same procedure is repeated with the RGPS data set starting from a 12.5 km resolution. At each step, the boxes of length $L$ are overlapping with their neighbors at their midpoint. The RGPS80 mask does not necessarily contain a whole number of boxes, $n \not\equiv 0 \mod \frac{L}{L_0}$ in general, where n is the maximal size of the mask along a given axis and $L_0$ is the resolution of one grid cell. The mean inside the fractions of squares that are left

at the boundaries of the domain is included only for boxes that are filled with at least 50% data. We calculate the deformations rates using the average in time and space velocities, and we also compute the effective size of the box by taking the square root of the total number of occupied cells in the box. From these points, we take the mean of the deformation rates for each box size and fit a least-square power law in the log–log space to find $\beta$, the spatial scaling exponent.

For the temporal scaling $\alpha$, we instead fix L to the spatial resolution value of the data set (10 km), and we compute the

mean deformations with the different time-averaged velocities ranging from 3 days to 24 days (i.e. $T = 3, 6, 12, 24$) and fit a least-square power law to calculate the temporal scaling exponent $\alpha$.

## 4.4 Multifractal Analysis

The scaling exponents ($\beta$ and $\alpha$) are functions of the moment $q$ of the deformation rate distribution:

$$\langle \dot{\varepsilon}_{\text{tot}}^q(L,T) \rangle \sim L^{-\beta(q)}, \tag{25}$$

$$\langle \dot{\varepsilon}_{\text{tot}}^q(L,T) \rangle \sim T^{-\alpha(q)}. \tag{26}$$

While it is usually assumed that the structure functions $\beta(q)$ and $\alpha(q)$ are quadratic in $q$ for the sea ice total deformation rates (Marsan et al., 2004; Bouillon and Rampal, 2015; Rampal et al., 2019), the structure functions are not necessarily quadratic in $q$ for the generalized multifractal formalism (see Schmitt et al., 1995; Lovejoy and Schertzer, 2007; Weiss, 2008; Bouchat and Tremblay, 2017), and are expressed instead as (for the spatial structure function),

$$\beta(q) = q(1-H) + K(q) = \frac{C_1}{\nu - 1}q^\nu + \left(1 - H - \frac{C_1}{\nu - 1}\right)q, \tag{27}$$

where

$$K(q) = \frac{C_1}{\nu - 1}(q^\nu - q). \tag{28}$$

In the above Equation, $C_1$ ($0 \leq C_1 \leq 2$) characterizes the sparseness of the field, $\nu$ ($0 \leq \nu \leq 2$, $\nu \neq 1$) is the Lévy index, or the degree of multifractality (0 for a mono-fractal process, 2 for a log-normal model with a maximal degree of multifractality),

and $H$ ($0 \leq H \leq 1$) is the Hurst exponent. We use a general non-linear least squares fit for the structure functions' parameters.





A similar equation holds for the temporal structure function $\alpha(q)$. $K(q)$ is called the "moment scaling function exponent" for a random variable. It defines the singularity spectrum, a function that describes the distribution of singularities (or points of non-smoothness) across different scales in the system.

Note that the scaling exponents for $q = 1$ ($\beta(1)$ and $\alpha(1)$) are equal to $1-H$, and therefore, a higher $H$ means a less localized
or smoother field. Moreover, the degree of multifractality $\nu$ defines how fast the fractality increases with larger singularities. As $\nu$ increases, larger deformation will dominate, and there will be fewer low-value smooth regions for example. $C_1$ represents how "far" the multifractal process is from the mean singularity value given by $\beta(1) = 1-H$; we can understand this by taking the derivative $\beta'(1) = (1 - H) + C_1$: the higher $C_1$ is compared to $1 - H$, the fewer field values will correspond to any given singularity, i.e., the singular field values are more sparsely grouped (Lovejoy and Schertzer, 2007).

As noted in Bouchat et al. (2022), the computed parameter values are sensitive to the number of points used to define the structure functions. Therefore, we use the same moment increments of 0.1 in order to derive the three multifractal parameters $(\nu, C_1, H)$.

## 5   Results

### 5.1   Simulated Total Deformation Field

In the control run ($d = 0$ or $n = \infty$), the simulated LKFs are more diffuse, less intense and the LKF density is lower when compared with RGPS observations (see Figure 2b). When including damage, LKFs are better defined, more intense, and the LKF density is higher, in better qualitative agreement with observations (this is true for all configurations of VPd models except $n = 1$); the ice strength along LKFs is much weaker, providing a strong positive feedback for the simulation of higher intensity and density of fracture lines, akin to RGPS-derived LKFs (see Figure 2). As $n$ decreases from $n = 50$ ($\sim$infinity) to $n = 1$, the
intensity, definition, and density of LKF increase until maximum damage is present in all grid cells and LKFs are no longer distinguishable from the undeformed ice, effectively rendering the ice soup-like 2. These results are robust to the exact choice of a healing timescale ($t_h = 2$–$30$ days), except when $t_h \approx t_d$ when fewer extreme deformation events are present. In all cases, however, the simulated LKFs are not as well-defined as the LKFs in RGPS observations presumably due to spatial resolution (see for instance Bouchat et al., 2022). Note that increasing shear strength ($e = 0.7$) with damage does improve the localization
of LKFs as for simulation without damage in accord with results from Bouchat and Tremblay (2017) (see Figure 2i). Another key visual difference is that the spatial mean of the deformation rates is higher for the VPd model than for the VP model and RGPS data, see also section 5.2 below for a discussion and more quantitative assessment.

The mean ice thickness over the Arctic Ocean is also sensitive to the amount of damage in the model (results not shown). For instance, the VPd model with $n = 5$ and $t_h = 2$ (low damage), and $n = 3$ and $t_h = 30$ (high damage) gives a 1 cm and 5 cm
mean ice thickness anomaly respectively. This thickness increase occurs mostly along LKFs in the form of ridges and clearly shows the impact of damage on the deformation fields. Interestingly, we see a reduction in sea ice thickness anomalies for the VPd model with maximal damage ($n = 1$ and $t_h = 30$). In this case, convergence (thickening) occurs over broader areas and when integrated, leads to a reduction in the positive ice thickness anomaly.





**Figure 2.** Simulated ($\mathrm{VPd}(e, n, t_h, P^*)$) and observed total deformation rates at a 10 km resolution (12.5 km for observations) for a 3-day average between January 29–31, 2002 compared with observations as a function of the ellipse aspect ratio ($e$), damage exponent ($n$), healing timescale ($t_h$, days), and compressive strength ($P^*$, kN/m$^2$). The VP with $e = 2$ (control) and $e = 0.7$ (VP(0.7)) are equivalent to $\mathrm{VPd}(2, 50, t_h, 27.5)$ and $\mathrm{VPd}(0.7, 50, t_h, 27.5)$ respectively.





## 5.2 Probability Density Functions (PDFs)

When considering damage, a larger number of LKFs is present for any mean total strain rate with a transfer from lower to larger total deformation rates in the PDF. This shift results in a linear decay in the tail of the PDFs (log–log plot) for shear rate and divergence/convergence that is in better agreement with RGPS. Interestingly, the VPd model is particularly good at reproducing the large divergence and convergence rate (and to a lesser extent large shear strain rate) present in RGPS observations contrary to the standard VP model that has a limited ability to simulate both observed divergence and convergence rate larger than

$10^{-1}\,\mathrm{day}^{-1}$ (see Figure 3). The PDFs of shear strain rates are more sensitive to the healing timescale $t_h$ than the damage exponent parameter $n$; with larger healing timescales leading to more shear. The best fit with observations occurs for $n = 3, 5$ and $t_h = 2$, or at $n = 1$ and $t_h = 30$. A smaller $n$ leads to more extensive but less intense damage that can be compensated by keeping a larger $t_h$. Similarly, the PDFs of convergence are more sensitive to $t_h$ than $n$, with larger values of $t_h$ resulting in more convergence. The best correspondences between models and observations are with no damage and a reduced ellipse

ratio ($e = 0.7$) or with low damage $n = 5$ with low healing timescale $t_h = 2$. Interestingly, higher values of $P^*$ with some damage have little to no impact on the convergence PDF contrary to lowering the ellipse ratio and to results from Bouchat and Tremblay (2017). Nevertheless, any damage configuration is better than the control run at reproducing high convergence events. In contrast, the PDFs of divergence are equally sensitive to $n$ and $t_h$ with more damage (lower $n$ or higher $t_h$) resulting in a higher count of large deformations in divergence. In this case, both configurations (VP(0.7) and VPd(0.7, 5, 30, 27.5)) with

a lower ellipse ratio ($e = 0.7$) overestimate divergence (Figure 3, yellow curves). Interestingly, a higher $P^*$ leads to higher divergence, in better agreement with observations (Figure 3, deep rose curves), with PDFs comparable to the fully damaged ($n = 1$) and lower ellipse ratio ($e = 0.7$) configurations.

We note that damage increases convergence and to a lesser extent divergence. This asymmetry between changes in positive and negative divergence, when damage is increased, precludes a perfect fit with observations with the default ellipse aspect

ratio. The fact that reducing $e$ from $e = 2$ to $e = 0.7$ or increasing $P^*$ both increase divergence while keeping convergence the same suggests that a combination of some damage ($n = 3, 5$, and $t_h = 2$) together with a higher $P^*$ or reduced ellipse aspect ratio will lead to the best fit in the three types of PDFs. See the section below on the sensitivity of the parameters for a nuanced discussion of their optimal values.



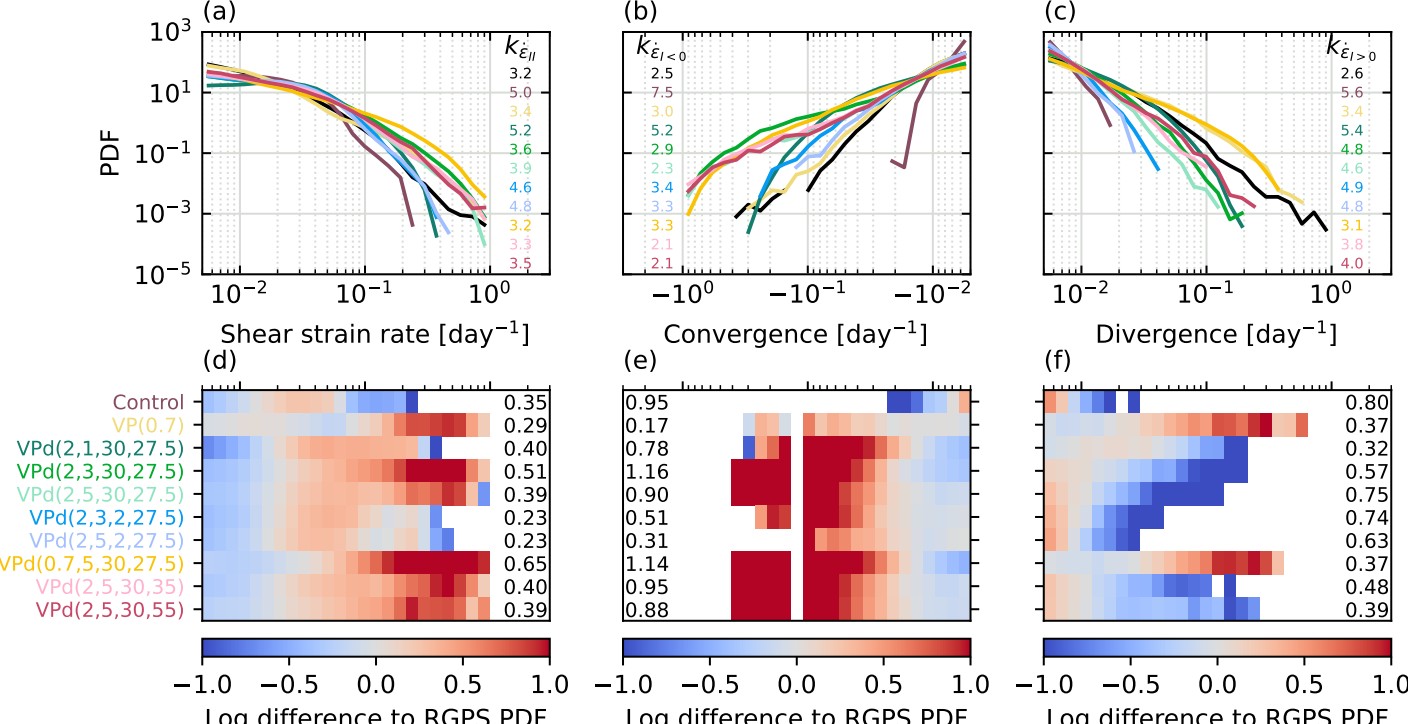

**Figure 3.** Top row: simulated (color) and observed (black) probability density functions for shear strain rate, convergence, and divergence at 10 km resolution and 3-day average ($L = 10$ km and $T = 3$ days) for January 2002. The power-law exponent calculated over one order of magnitude from the end of the distributions for each model and RGPS are shown in the inserts. Bottom row: binwise difference between the logarithms of models and RGPS PDFs. The average absolute difference per bin is shown in the inserts.

## 5.3 Cumulative Density Functions (CDFs)

The cumulative density functions (CDFs) (Figure 4) of the two models differ substantially because of the higher count of large deformations of the VPd model bringing its CDFs further from that of the control run. For shear strain rate, the KS-distances computed from the CDFs of the different configurations of the VPd model are all slightly higher ($0.21 \leq D_{\dot{\varepsilon}_{II}} \leq 0.36$) than that of the control run (0.19). The fact that the latter crosses the CDF of the data while keeping a similar maximal vertical range as the CDFs of the VPd model results in this slightly lower KS-distance, something that is not apparent from the PDFs

alone. In contrast, the KS-distances of the VPd CDFs for convergence are similar or smaller ($0.07 \leq D_{\dot{\varepsilon}_{I<0}} \leq 0.40$) than that of the control run (0.37). Not surprisingly, the configurations with $t_h = 2$ have a very low KS-distance (0.07 and 0.10), in line with the PDF of convergence that showed that large values of $t_h$ result in overshooting. Yet again, the key improvement resides in the divergence rate with KS-distances for the VPd model configurations that are smaller ($0.05 \leq D_{\dot{\varepsilon}_{I>0}} \leq 0.43$) than that of the control run (0.53), highlighting the success of the VPd model at simulating a higher count of large deformations



in divergence. Again, VPd configurations with $t_h = 2$ days have the largest KS-distance in divergence with values closer to the control run (0.36 and 0.43). Interestingly, the best fit with observations comes from the standard VP model with a reduced ellipse aspect ratio ($e = 0.7$) with very small KS-distances (0.03, 0.03, 0.15 respectively). These small values may be due to the interannual variability in the RGPS data; the KS-distances of a particular RGPS year can vary by as much as 0.17 when compared to the RGPS mean (Bouchat et al., 2022). Nonetheless, combining damage ($n = 5$, $t_h = 30$) with an increased $P^*$ does lead to very small KS-distances (respectively, 0.21, 0.20, and 0.20) and supports the conclusions drawn from the PDFs alone. Unsurprisingly, the KS-distance decreases with increasing $n$ and decreasing $t_h$ for shear strain rate and convergence, while for divergence, the KS-distance decreases with decreasing $n$ and increasing $t_h$ — as for the PDFs.

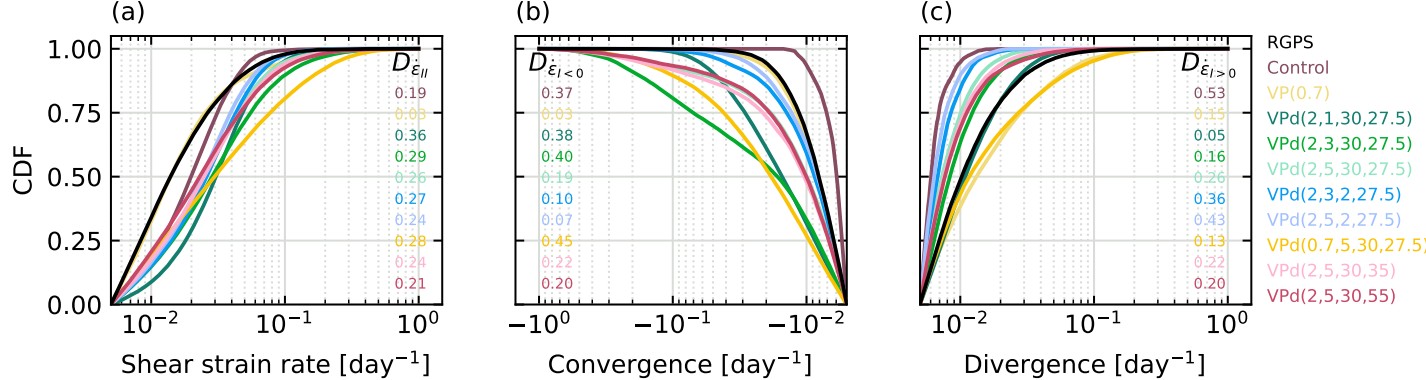

**Figure 4.** Simulated (color) and observed (black) cumulative density functions for shear strain rate, convergence, and divergence for models at 10 km resolution ($L = 10$ km and $T = 3$ days) for January 2002. The Kolmogorov-Smirnov distance between each model and the CDFs of RGPS observations is shown in the inserts.

## 5.4 Spatiotemporal Scaling

Both the VPd and VP models are able to reproduce some level of spatial and temporal scaling, as in RGPS (Figure 5-6). The spatial scaling exponent $\beta$ at $T = 3$ days of the VPd model is highly sensitive to the exponent $n$ and the healing timescale $t_h$; it increases with decreasing $n$ and increasing $t_h$, i.e. with more damage. The spatial scaling exponents are ranging from $\beta = 0.06$ to $\beta = 0.14$ for the different configurations of the VPd model, with the slope of the spatial scaling curve for the fully damaged VPd($2, 1, 30, 27.5$) model being morally the same as that of RGPS (0.15), while the standard VP model has a 3 times smaller exponent ($\beta = 0.05$); all configurations of the VPd model have better spatial scaling than the VP model. Note how reducing the ellipse ratio ($e = 0.7$, as proposed by Bouchat and Tremblay, 2017) also increases the spatial scaling exponent for the VPd model (yellow curve). The increase in the scaling factor for the VPd model indicates that LKFs are more localized in space than those of the VP model.





On the other hand, the temporal scaling $\alpha$ at $L = 10$ km of the VPd model for all configurations is lower ($\alpha = 0.13$ to $\alpha = 0.19$) than that of the observations (0.28) or the VP model (0.23). Note that the combination of damage and a reduced

ellipse aspect ratio ($e = 0.7$) decreases the temporal scaling exponent (yellow curve), contrary to its effect on the spatial scaling exponent.

Interestingly, all VPd simulation curves have a higher mean deformation rate (for both the spatial and temporal scaling), since damage increases the mean velocity of the ice (result not shown). Increasing $P^*$ reduces the mean ice velocity and the mean deformation rates across all scales to the same level as the control run (deep rose curves compared to light green curves).

This shift towards higher mean deformations is visible from the pan-Arctic simulations but has no impact on the spatial and temporal scaling.

In summary, the VPd model improves spatial localization at the expense of a weaker temporal localization of deformations. Temporal localization (or scaling) is not to be confused with intermittency. Temporal localization originates from the autocorrelations of the deformations time series at a given location and the rate at which these correlations decrease when increasing

the time lag between deformation rate values. In other words, a lower temporal scaling means that a high deformation event is more likely to be followed by another high deformation event in the "near future", resulting in a smeared time localization in the mean at a given scale. On the other hand, intermittency (or heterogeneity) is reflected in the *change* of localization within the same data set; the intermittency can be quantified from the shape of the structure function (as discussed below in section 5.5). With this in mind, it is expected that the VPd model would have a lower temporal scaling, as the damage increases the

probability of future (for $t < t_h$) deformation at a given grid cell. For the same reason, decreasing $t_h$ increases temporal scaling.



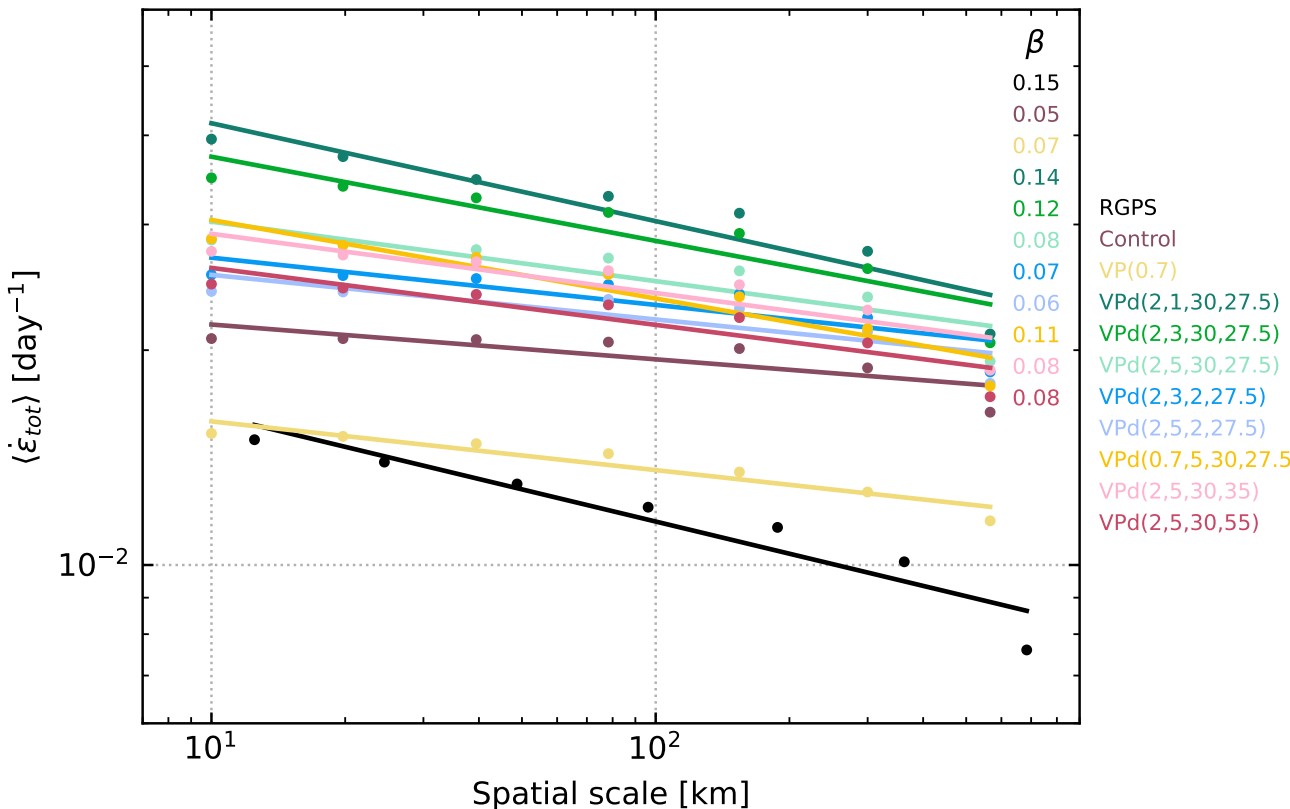

**Figure 5.** Simulated (color) and observed (black) spatial scaling of mean total deformation rates for $T = 3$ days in January 2002. Lines are least-square power-law fits, and their slope gives the scaling exponent $\beta$ (shown in the insert).



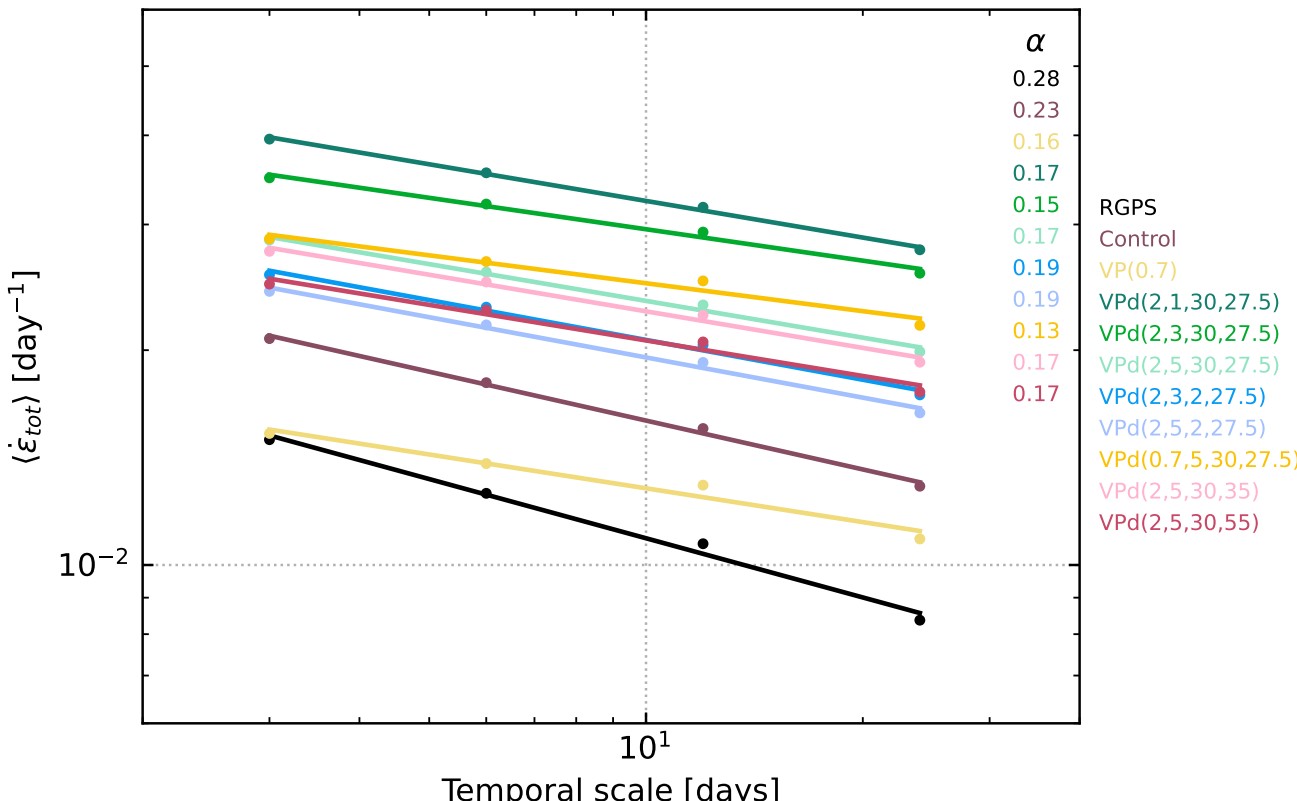

**Figure 6.** Simulated (color) and observed (black) temporal scaling of mean total deformation rates for $L = 10$ km in January 2002. Lines are least-square power-law fits, and their slope gives the scaling exponent $\alpha$ (shown in the insert).

## 5.5 Multifractal Analysis

When fractal structures have local variations in fractal dimension, they are said to be multifractals. In the case of sea ice deformation or strain rates, multifractality arises from the higher space and time localization of larger deformation rates, compared to smaller deformations (Weiss and Dansereau, 2017; Rampal et al., 2019).

The spatial structure functions of all the VPd configurations are in better agreement with observations when compared with that of the control run (Figure 7). The spatial multifractality parameter ($1.50 \leq \nu \leq 1.96$) of the VPd configurations increases when increasing $t_h$, but the dependence on $n$ only appears for high values of $t_h$. Larger values of $\nu$ characterize a field dominated by singularities of larger values; for sea ice, this means that configurations of the VPd model with a small healing timescale reflect this poorer multifractal behavior because the sea ice heals faster. For short healing timescales ($t_h \approx 2$) the

dependency of the multifractal parameter $\nu$ on $n$ disappears, but for $t_h = 30$, the dependency of $\nu$ on $n$ becomes apparent;





the spatial multifractality parameter $\nu$ reaches a local minimum ($\nu = 1.61$) for $n = 3$, followed by a local maximum at $n = 5$ ($\nu = 1.96$), then plateaus at some intermediate value ($\nu = 1.76$) as damage decreases towards that of the control run (see insert of Figure 7).

The VPd$(2, 3, 30, 27.5)$ configuration highlights a complex transient state in the multifractal behavior of the model from fully
damaged ice (the VPd$(2, 1, 30, 27.5)$ configuration) with high multifractality ($\nu = 1.94$) but low heterogeneity ($C_1 = 0.04$), to high multifractality ($\nu = 1.96$) and high heterogeneity ($C_1 = 0.14$) corresponding to the VPd$(2, 5, 30, 27.5)$ configuration. Further decreasing damage (e.g. VPd$(2, 50, t_h, 27.5)$) leads to lower values of both multifractality and heterogeneity. The heterogeneity of the field ($C_1$) of all VPd model configurations ($0.04 \leq C_1 \leq 0.21$) are also in better agreement with observations ($C_1 = 0.17$) than that of the control run ($C_1 = 0.03$) although still lower than RGPS for the lower values of $t_h$ and $n$, again
suggesting that the VPd model is better at focusing LKFs spatially. This is also in agreement with the higher Hurst exponent for the control run ($H = 0.95$) suggesting a spatially smoother field than the different configurations of the VPd model ($0.85 \leq H \leq 94$) and RGPS observations ($H = 0.87$). This is, again, consistent with the results from the spatial scaling analysis. Interestingly, values of the Hurst exponent at $q = 1$ do not necessarily translate into having observation-fitting values in the other two multifractal parameters, which leads to graphs that are far from that of RGPS observations. Notably, the
VPd$(2, 1, 30, 27.5)$ has a similar value for the Hurst exponent ($H = 0.86$) compared to RGPS observations ($H = 0.87$), but has the lowest heterogeneity ($C_1 = 0.04$) of all the VPd model configurations, resulting in one of the poorest representation of the observations, together with the $t_h = 2$ configurations.

The most striking differences between the control run and the VPd model are their heterogeneity and spatial autocorrelations. Combining the damage parametrization with a different value for the ellipse ratio ($e = 0.7$) further increases the heterogeneity
($C_1 = 0.21$) of the deformation field at the cost of lowering the spatial multifractality ($\nu = 1.57$). Increasing $P^*$ also leads to higher heterogeneity ($C_1 = 0.16$), while still maintaining the high values of the multifractality ($\nu = 1.95$). Interestingly, a third root appears in the range $q < 1$ when we change the ellipse aspect ratio or $P^*$ (see Figure 7). The multifractal theory does not allow for more than two roots, and the fact that this is observed is indicative that the model (the VPd at least) might not follow the multifractal theory. This might also be the case for the other configurations, the observations, and the control run. Whether
this is a new behavior associated with the damage parametrization in tandem with the change in ellipse aspect ratio and $P^*$ or an enhancement of an already existing property remains to be investigated.

The differences in the temporal structure functions of the VPd model and the control run are more subtle (see Figure 8). Temporal multifractality is also reproduced by the different configurations of the VPd model ($1.20 \leq \nu \leq 1.86$), and they are all somewhat worse than the standard VP model ($\nu = 1.67$) compared to RGPS data ($\nu = 1.87$). Similarly to the spatial structure
functions, almost all configurations of the VPd model are as temporally heterogeneous ($0.04 \leq C_1 \leq 0.22$) — also called intermittency — as the observations ($C_1 = 0.14$), while the control run is the least heterogeneous ($C_1 = 0.09$), except for the fully damaged VPd$(2, 1, 30, 27.5)$ configuration. RGPS observations have a somewhat low Hurst exponent value ($H = 0.73$), while all configurations of the VPd model have a high value ($0.82 \leq H \leq 84$), even compared to the control run ($H = 0.77$). This high Hurst exponent brings down the graph of the VPd temporal structure functions, even if their curvature (governed by
$\nu$ and $C_1$) is always higher than that of the control run structure function, and in agreement with the curvature of the graph




of the structure function computed from RGPS observations, especially for high values of $n$ and $t_h$. This curvature change accounts for the majority of the difference between the simulated temporal structure functions and the high Hurst exponent is indicative of a temporally smoother field — in agreement with the results from the temporal scaling analysis. The only configuration that has a lower curvature than the control run is the fully damaged VPd$(2, 1, 30, 27.5)$. This configuration has

both low heterogeneity and high Hurst exponent, leading to a temporal structure function that does not have enough curvature. Overall, reducing $n$ (more damage) reduces the temporal multifractality, and reducing $t_h$ reduces the heterogeneity. Moreover, increasing $P^*$ or reducing the ellipse ratio increases heterogeneity but reduces multifractality. Interestingly, the Hurst exponent is almost constant for all configurations of the VPd and the standard VP with a reduced ellipse aspect ratio. As in the spatial structure function, changing the shape or size of the ellipse does unveil a third root in the temporal structure function in the

range $q < 1$, which is indicative that the VPd model does not follow the multifractal theory.

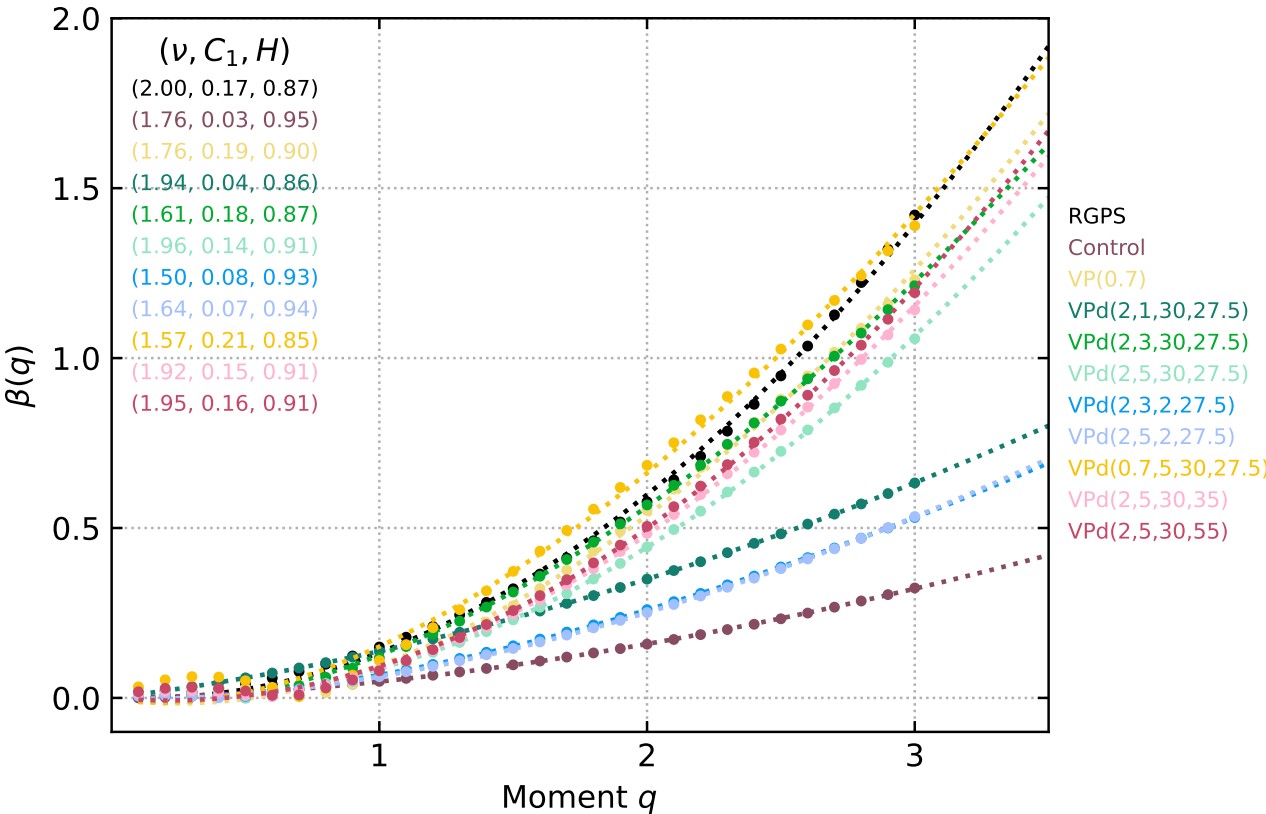

**Figure 7.** Simulated (color) and observed (black) spatial structure functions $\beta(q)$ of the total deformation rates for $T = 3$ days for January 2002. Dotted lines are the least-square fit for Equation 27, and the inserts are the value of the parameters of the fit $(\nu, C_1, H)$.



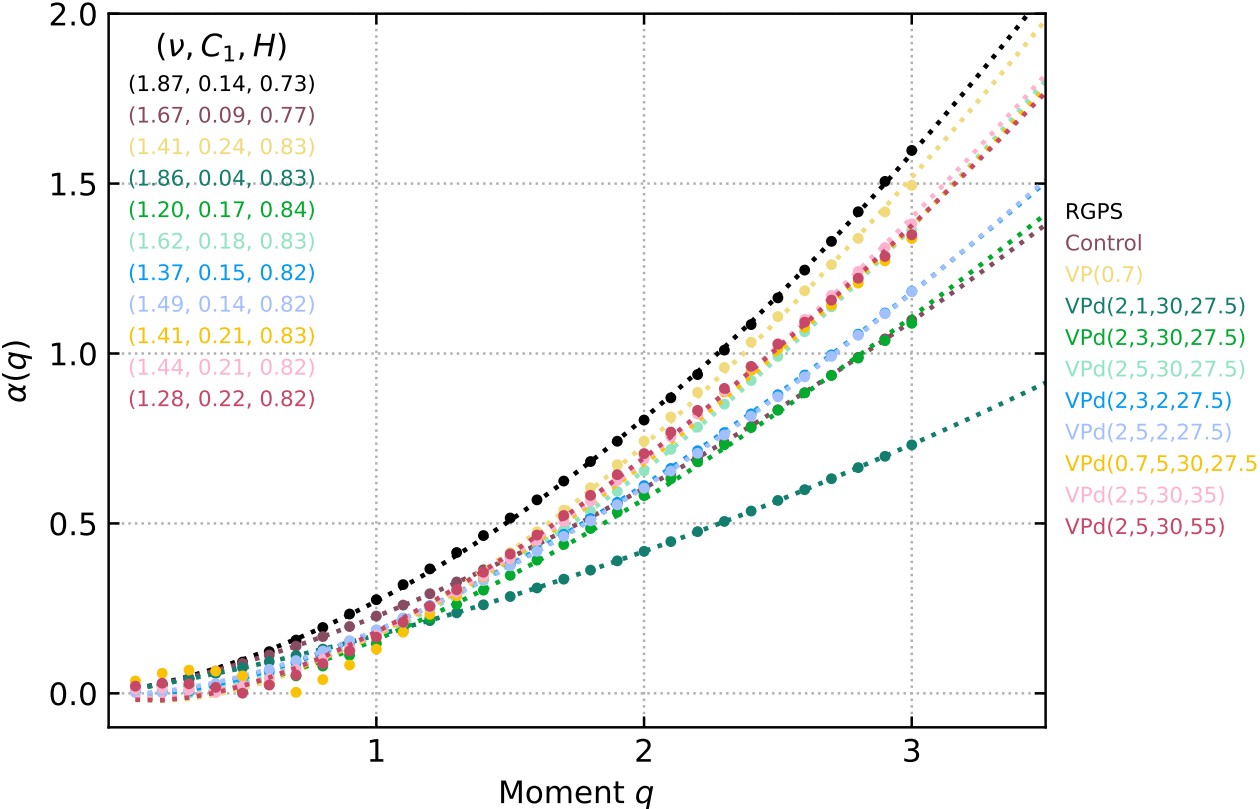

**Figure 8.** Simulated (color) and observed (black) temporal structure functions $\alpha(q)$ of the total deformation rates for $L = 10$ km for January 2002. Dotted lines are the least-square fit for Equation 27, and the inserts are the value of the parameters of the fit ($\nu, C_1, H$).

### 5.6 Sensitivity to $t_h$, $n$, and the $e$

In the VPd model, a shorter healing timescale results in an overall smoother deformation field with fewer intense LKFs (see Figure 2e–h). Therefore a shorter healing timescale in this model is not necessarily wanted, as it reduces the effects of the damage source term (see Equation 16). As a result, the spatial scaling improves marginally, but the temporal scaling becomes significantly worse (see blue curves and their insert in Figures 5 and 6). This is also apparent in the multifractality as there are only small discrepancies between the VPd model with a short healing timescale and the control run (see Figures 7–8). The optimal healing timescale value $t_h^*$ therefore should be on the order of one month rather than days in a VPd model, in contrast with the value commonly used in the MEB model of 1 day and that derived from observations (Dansereau et al., 2016; Murdza et al., 2022). This is of course expected since damage in the VPd model does not represent necessarily the same thing




as damage in the MEB model. Moreover, in Murdza et al. (2022), the authors raised the question of whether the rapid strength recovery of the ice that they measured can be applied to larger scales.

In the VPd model, deformation rates are sensitive to the exponent parameter $n$. When $n$ is low, the damage reaches one in a few time steps, and remains high, such that all the ice is nearly fully damaged (see Figure 2d), except for grid cells in the viscous regime. When $n$ is large ($> 50$), the VPd model gives morally the same results as the VP model. Considering all

deformation metrics above, we suggest the value of $n^* = 5$ for the damage parameter $n$.

When combining these values with the reduced value for the ellipse ratio ($e = 0.7$ Bouchat and Tremblay, 2017), we find that the spatial scaling is stronger, while temporal scaling is even lower. This is in disagreement with Bouchat and Tremblay (2017) who found that changing $e$ increases both spatial and temporal scaling. This is presumably due to the fact that reducing $e$ strengthens the ice in shear, and thus enhances the impact of the damage parametrization. Moreover, increasing $P^*$ does

result in better multifractality and magnitude of deformation rates, without any consequences on the scaling. We suggest to increase $P^*$ when implementing the VPd model.

## 6 Discussion

Deformation rate statistics simulated by the VPd model are in better agreement with RGPS observations and than that of the standard VP model. Not surprisingly, the plastic rheology with damage is particularly good at reproducing the spatial scaling

and structure function. Moreover, while a lower temporal scaling was achieved with the damage parametrization, the temporal intermittency of the VPd model was slightly higher and closer to the observations. This shows that the inclusion of a damage parametrization inside a model has a non-negligible impact on the scaling, multifractality, and heterogeneity of the deformation fields both spatially and temporally.

Considering that the VP model can still produce some low level of multifractality, we hypothesize that the governing factor

in reproducing deformation rate statistics is not necessarily the physics behind the parametrizations nor the pre-fracture elastic regime but rather the "amount of memory" of past deformation present in a model. Memory in the VP model is present through the concentration and thickness of the ice; in the VPd model (or EB family), memory is also associated with damage which is present for both convergent and divergent flows and has a much faster timescale ($t_d = 1$ day) than $h$ and $A$. Another possibility could simply be the addition of some form of spatiotemporal heterogeneity in the ice strength, which the damage parametriza-

tion presented in this study does — highlighting that even ad-hoc parametrizations are going to improve deformation rate statistics.

Since damage is expressed in terms of the bulk viscosity term, the "memory" of the system resides in the ice strength through the damage coupling factor (see Equation 18). The plastic deformation therefore instantaneously reduces the ice strength locally. This new memory in the system complements the memory associated with sea ice divergence via the concentration

and thickness of the ice. That is, the ice is more susceptible to break where — or near where — it has been previously broken. LKFs are, therefore, a memory network of the viscous-plastic model that includes a damage parametrization with a "learning" curve that depends on the specific choice of damage timescale and exponent with a slow regenerative healing mechanism that





acts as a memory eraser. This behavior is reflected in higher temporal intermittency as well as a higher spatial multifractality, heterogeneity and scaling in the VPd model. The downside is that the temporal multifractality and scaling exponent in the VPd

model are lower, which indicates that long-time autocorrelations are especially strong in the VPd model. This is explained by the memory of previously damaged ice, which prompts the ice to break where it already broke in the past.

Usually, when critical stress is reached in an MEB model, the Young's modulus is instantaneously reduced locally, and the excess stress results in brittle fracture and increased damage. On the other hand, in a standard viscous-plastic model, when plasticity is reached, the ice strength is reduced only for large — grid-scale — diverging ice events. In this scenario, the ice

thickness and concentration are reduced, leading to a lower ice strength at the next time step. This process is slow and much smoother than the one in the VPd model, which mimics the behavior of the MEB model. In that regard, the VPd model permits new types of weakening that reduce the ice strength (i.e., shear and convergence), something that is not possible in a standard VP model, hence creating more well-defined LKFs that lead to a better statistical fit of the observations. This is reflected in the higher counts of high deformation events in both convergence and divergence.

In the VPd model with a modified smaller ellipse aspect ratio, a third root appeared in both the spatial and temporal multifractality plots. This means that the theory, which is only valid for a Lévy index between 0 and 2, does not hold anymore. Is this particular configuration of the VPd model uncovering a new property, or is it simply amplifying something that was already there, and was overlooked? What does it mean for the multifractality of LKFs?

In light of the results presented above, we recommend the implementation of this damage parametrization in a standard

viscous-plastic model. This parametrization comes at no additional cost, contrary to increasing the spatial resolution of the model, which increases the computational time of simulations by a factor of ∼25 for a 5-fold increased spatial grid resolution of 2 km × 2 km, or even the tuning of the ellipse ratio, which decreases the numerical convergence substantially. The damage parametrization, together with a careful choice of yield curve parameters (see for example Bouchat and Tremblay, 2017; Bouchat et al., 2022) would prove to be a low-cost, efficient way of improving deformation statistics, even if sea ice models

are not run a very high resolution.

As the MEB model includes a damage parametrization, we ask the question of whether the agreement between the MEB model and the RGPS observations is in part due to this sub-grid fracturing parametrization in conjunction with the Lagrangian mesh used in MEB models, rather than the explicit choice of rheology — elastic deformation followed by brittle fracture. Recent studies (together with results presented here) suggest that the inclusion of a damage parameter (Plante et al., 2020) and

the Lagrangian mesh (Bouchat et al., 2022) are key factors in a better description of deformation rate statistics. RGPS observations are obtained from the displacement of tracers at a 10 km spatial scale, but ice motion is much more complex, and these observations of emergent properties include the effects of processes that take place at much finer scales (sub-kilometer) such as bending, twisting, micro-fractures, and fusion. We hypothesize that efforts put into developing sub-grid parametrizations will be the go-to for fast and light deformation rate statistics improvement in the short term. Notably, using discrete element models

(DEM) as toy models for developing and calibrating new sub-grid-scale parametrizations may provide exciting results.

Note that we used the same methodology as in Bouchat and Tremblay (2017). This is important to keep in mind as their results show that maximum likelihood estimators (MLE) of the scaling parameters for the tail of PDFs of RGPS gridded




deformation products are 29% (convergence), 25% (divergence), and 14% (shear) higher than those obtained using RGPS Lagrangian product (Marsan et al., 2004; Girard et al., 2009). They attributed about 10% of the higher scaling parameters to the choice of mask and the rest to the smoothing inherent to the gridding procedure. Therefore, our results are not necessarily reflecting reality, but nevertheless are still useful as they help discriminate our model's configurations with RGPS gridded observations for a particular year. The results presented are robust to the exact choice of year. However, the mask we are using is located above the Canada Basin and extends to the East Siberian Sea, and we are only using the data from January 2002. Exact numbers are therefore probably influenced by local — in space and time — effects. As a matter of fact, when doing the same analysis for other years, the values for the parameters of the multifractal analysis and the PDFs decay exponents vary, but conclusions drawn from this study are robust, as the general behavior of the models stays the same for different years (results not shown). It is believed that specific numbers given here are not necessarily representative of reality, but are rather just a rough estimate of the behavior of the models and RGPS.

## 7 Concluding Remarks

We implement a sub-grid damage parametrization in the standard viscous-plastic model to investigate the effects of damage on the deformation rate statistics, namely, the probability density functions (PDFs) exponential decay and shape, the Kolmogorov-Smirnov distance between cumulative density functions (CDFs) of simulations and observations, the spatiotemporal scaling exponents, and the multifractal parameters expressing the spatiotemporal structure functions. Results show that the deformation rate statistics are very sensitive to the inclusion of a damage parametrization, including advection of damage and a healing mechanism. Therefore, we argue that sub-grid-scale parametrizations should be considered when comparing different rheological models. Specifically, we find that this new damage parametrization improves power-law scaling and multifractality of deformations in space in the viscous-plastic model, the trade-off being a lower exponent than the standard VP model for the temporal power-law scaling. We show that the new VPd model increases the number of large divergence and convergence rates in better agreement with RGPS observations as per the new quantitative metric introduced by Bouchat et al. (2022). Moreover, we show that the VPd model is especially good at producing spatial multifractality, which was expected since the damage parameter was constructed to improve the spatial localization of LKFs. The fact that the standard VP model can still produce some spatial multifractality, without including any "cascade-like" mechanisms that would permit multifractality as in the VPd model, indicates that other physical mechanisms are at play in both models. These other mechanisms are not identified, and the origin of multifractality in the VP model remains an open question. We hypothesize that one likely candidate is the "amount of memory" that a model possesses. The proposed damage parametrization is a compelling low-cost add-on to viscous-plastic models

The implementation of the proposed damage parametrization inside viscous-plastic models provides an efficient, low-cost option for improving deformation rate statistics in low-resolution sea ice models, in tandem with a relatively long healing timescale and an increased $P^*$. Other possibilities would be to couple the damage parameter to the ellipse ratio directly rather than the ice strength, which would change the physics of the ice locally rather than changing its strength. Future work will



include other sub-grid scale parametrizations, such as the inclusion of memory through an evolution equation for dilation along Linear Kinematic Features — memory seems to be a determining factor for deformation statistics — and non-normal flow rules, i.e. rheologies that allow for plastic deformations and for time-varying internal angle of friction. These would allow models to have a better memory of past deformations.

570 *Code and data availability.* All analysis codes are available on GitHub: https://github.com/antoinesavard/SIM-plots.git. All published code and data products can be found on Zenodo: will.be.put.at.final.submission. This includes the published analysis code (**?**), the ice velocities from model output (**?**), and RGPS gridded velocity derivatives (Kwok, 1997).

*Author contributions.* AS and BT designed the experiments and AS carried them out; AS developed the model parametrization and statistical analysis code and performed the simulations; AS analyzed the models' outputs; AS wrote the manuscript draft with contributions and reviews 575 from BT; AS archived the code and the models' outputs.

*Competing interests.* The authors declare that they have no conflict of interest.

*Acknowledgements.* A. Savard is grateful for the financial support by Fond de Recherche du Québec – Nature et Technologies (FRQNT), ArcTrain Canada, McGill University, and the Wolfe Chair in Scientific and Technological Literacy (Wolfe Fellowship). B. Tremblay is funded by the Natural Sciences and Engineering and Research Council (NSERC) Discovery Program, and by Environment and Climate 580 Change Canada (ECCC) via the Grants and Contributions program. We also thank Québec-Océan for financial support.



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
