# Peer review of "Damaging viscous-plastic sea ice"

_EGUsphere, 2023_

## Referee Comment (RC1)

In this manuscript, the authors propose to introduce in a viscous-plastic (VP) sea ice model a "memory" parameter that they call (inappropriately, see below) "damage", making the ice weaker (i.e. its strength $P$, setting the size of the yield curve, decreases) each time it yields. This generates a positive feedback leading to a weaker and weaker ice, an effect counterbalanced in their model by "healing" (eq. 15). The authors implement this scheme into their VP model and run 1-month simulations over the Arctic forced by geostrophic wind fields. They compare the results of their model with "classical" VP simulations (i.e. without this memory effect) in terms of strain localization and scaling, and argue that this new parameterization improves the modelling of strain localization (which makes sense considering the positive feedback mentioned above) and spatial scaling, but not the temporal scaling.

I have little comments about the numerical scheme, the forcing, etc… I have some about the evaluation and the comparison with RGPS data (see below). My main concern, however, is about the *concepts*. The authors define their memory parameter as "damage", and argue that it is equivalent to the damage parametrization in the EB/MEB family of models. Since the seminal paper of [*L M Kachanov*, 1958], damage mechanics became an actual branch of solid mechanics with (at least) one international journal devoted to it, many classical works (e.g. [*M Kachanov*, 1994; *Kondo et al.*, 2007] among many others) and several books (e.g. see [*Lemaitre and Chaboche*, 1990]). In all these works, damage refers, in continuum mechanics, to a degradation of *elastic* properties as the result of emerging internal defects (microcracks, voids,..), an effect that is homogenized at the considered scale. Note that damage (with this true definition) can also be extended to granular media [*Karimi et al.*, 2019], without the need to break bonds or particles, just from topological rearrangements between particles, that effectively lead to *elastic* softening. The memory parameter introduced by the authors has therefore nothing to do with damage, which, by construction, cannot be distinguished implemented in a model that ignores elasticity. This is not just a matter of semantics: See e.g. L61-65 "Damage parametrizations — first developed in rock mechanics — are ad-hoc in that they are not derived from observations and/or from first physics principle". This is a very strong, yet false statement. Damage has actually been formulated from first physical principles, observed, measured in many communities (mechanics, geomechanics, geophysics (e.g. from a decrease of elastic wave velocities around major faults after an earthquake [*Brenguier et al.*, 2008]), etc, see the literature suggested above). Consequently, in its present form, this manuscript would falsely perpetuate the idea within the sea ice and perhaps other communities that damage is an ''ad-hoc'' parameterization, and doing so, would ignore only about 65 years of work.

On the reverse, the memory effect introduced here is indeed an ad-hoc parameterization. Note that in the MEB framework, the dependence of the viscosity on damage is partly empirical as well, within some physical bounds.

In the EB/MEB modelling framework, damage was introduced with two objectives:
- introduce indeed a memory of past loading history. The difference with the present work is that the elastic properties are evolving through time, i.e. following damage mechanics, not the strength.
- even more importantly, the objective was to propagate elastic stresses within the elastic medium following a damage/fracturing local event, this way allowing fracturing itself to propagate in an appropriate way. This comes back to the historical work of [*Eshelby*, 1957]: an inclusion within an elastic body, with elastic constants within the inclusion different from the surrounding medium, generates an elastic stress redistribution kernel. This is exactly what is modelled within the EB/MEB framework, and obviously not within the VP framework, with or

without a memory parametrization. This fundamental difference should be clearly stated in the manuscript to avoid confusion.

Still on concepts: to imagine that a material previously damaged will yield/fail more easily than an intact material seems to make sense at first glance. However, the relation between damage and strength is far to be trivial, certainly non-linear, most likely strongly dependent on the material, the loading geometry ect… Actually, progressive damage models have been used to explore this complex physics, and showed that strength generally results from complex (including long-ranged elastic) interactions between defects, resulting in non-trivial size effects [*Weiss et al.*, 2014]. Even if ignoring the interactions between defects, the ultimate strength of a solid will depend on the (extreme) statistics of the population of defects, actually on the "largest" flaw in the [*Weibull*, 1939] classical approach, and not on homogenized "damage".

Regarding the parameterization of memory implemented in this paper, it also poses some contradictions with the VP constitutive equation. One way to understand this contradiction and the incompatibility of the parameterization with actual damage is that the authors implement a time-evolving "damage" evolution equation (with a given propagation time scale). Damage, by definition, measures the effective stress and its propagation is therefore an image of stress redistribution. However, the viscous(-plastic) constitutive equation intrinsically assumes a *steady state* between stress and deformation rate. In other words, it does not account for stress redistribution: there is no $d\sigma/dt$ (propagation, or elastic) term, only a $\sigma$ (diffusion, or viscous) term.

On the more specific question of time scales, the authors propose a time scale for the propagation of "damage" without justifying it on a physical basis, or specifying the associated space scale considered. This is missing from the paper and has repercussion on the interpretation of the appropriate, nominal healing time scale (see specific comments below). As in previous works on visco-elastic-brittle-type models, the conclusions of the authors however support the requirement of a clear separation between the damaging and healing time scales. The fundamental underlying contradiction arising from the absence of such separation (healing almost as fast as fracturing) should be introduced much earlier in the text.

Concerning parts 4 and 5 and the comparison of model results to RGPS data in terms of strain localization as well as space and time scaling:

- I do not understand the interest to compare models outputs to observation in terms of PDFs (5.2) and then CDFs (5.3). The very same information is contained in both.

- On figure 4, the data seem to have been plotted (or accumulated) in a wrong way, i.e. $P(<X)$ is shown, not $P(>X)$. This should not be done that way, as small strain-rate values (at least, in the observed dataset) suffer from signal/noise biases, so we are more interested by the upper tail of the distribution. Consequently, I do not understand the associated evaluation (section 5.3).

- Overall, the authors report some "scaling", or at least a scale-dependence of strain-rates. This, in my sense, is not completely enough to argue that the model incorporates the right physics (or not). Indeed, the models (VP or VPd) are here forced using geostrophic wind fields which are themselves characterized by some complexity. And so spatial gradients in the forcing fields will inevitably induce some strain localization. The memory effect introduced here seems to reinforce this, which appears reasonable indeed. A real test would be to force such model with

a homogeneous field. In that case, spatial scaling naturally emerges using progressive damage models, as the result of elastic long-ranged interactions and threshold mechanics (so, the introduction of damage).

Nevertheless, some important differences between sea ice scaling properties and fluid turbulence exist, such as a space/time scaling symmetry, which is not discussed here. See e.g. [*Weiss*, 2013] for a discussion on this topic.

In conclusion, to the question "Can we introduce a memory effect in VP sea-ice models from an empirical parametrization of the strength evolution?", I would say, why not ? , and it could indeed improve, empirically, the representation of sea ice deformation fields to some extent. However, the manuscript in its present form cannot be recommended for publication, for all the conceptual reasons given above. First, the term "damage" should be removed everywhere from the text, including in the title, as this manuscript is not dealing with this concept (see above). Then, a clear distinction between the concepts used here and in EB/MEB should be stated clearly, following the comments above.

**Specific comments**

Abstract: ''We implement a damage parametrization in the standard viscous-plastic sea ice model to disentangle its effect from model physics (visco-elastic or elasto-brittle vs. visco-plastic) ''. This sentence is misleading. First, how is damage not part of the model physics? Second, you do not compare a visco-plastic model with a visco-elastic model here that includes or not damage. Please rephrase.

Line 21: "Internal stresses rapidly redistribute these forces from ice–ice interactions over long distances.'' This is exactly what an elasto-brittle model (based on elasticity and progressive damage) is built to represent, and what a VP model does not consider, either with or without a memory effect included.

Line 35: ''sea ice motion''. And deformation.

Lines 36-40: ''sea ice is considered as a highly-viscous fluid *for small deformations*. In this case, sea ice deforms as a creeping material.'' For small *stresses''*, instead. Please clarify.

Line 48: ''the inclusion of deformation on discontinuities'': not clear. What do you mean? Deformation or discontinuities should not be introduced in an ad-hoc manner, but result from the modelled rheology.

Line 49: "anisotropic yield curve *that allows* tensile stresses".

Line 51: "Finite element models…". This should be taken as another main comment: Another source of confusion in the paper is the association of certain rheologies (EB-type, Elastic-decohesive) but not others (VP) to the numerical schemes that have been employed in models implementing these rheologies. This gives the wrong impression that there is some causal relation between the numerics and the model equations. These are actually two distinctive things: you first formulate a system of equations and if consistent, it is independent from the choice of spatio-temporal discretization scheme you then apply. In the discussion, you even extend your supposition of good/bad agreement with observations to the Eulerian vs Lagrangian framework in which the equations are cast. However, as for numerical schemes, you never offer

and explanation as to why. These suppositions add up, so much so that the reader is lost: what is best between damage or not damage, memory or not, rheology vs numerical scheme. The comparison you make here allows distinguishing only between a VP model with and without a memory effect and that should be made clear.

Line 53: ..''that damage associated with (prior) fractures also affects ice strength''. No ! You are here making a confusion between elastic modulus and strength (stress at failure). If both are expressed in Pa, they are completely different concepts – see also general comments above.

Line 61-65. Please remove this wrong statement (see above).

Line 65: ''the damage is expressed as a function of the stress overshoot''. Not clear: The (temporal) evolution of damage, or damage *increment*, is defined as a function of the stress overshoot.

Lines: 69-70: Again, you introduce some confusion between the definition of damage itself and its incrementation or propagation, or between *damage* and *fracture* mechanics. Please clarify.

Lines 72-74: '' Earlier model–observation comparison studies, aimed at defining the most appropriate rheology for sea ice, found that any rheological model that includes compressive and shear strength reproduces observed sea ice drift, thickness, and concentration equally well (e.g. Flato and Hibler, 1992; Kreyscher et al., 2000; Ungermann et al., 2017)''. It seems a little odd to cite here these early studies of rheological model comparisons, which do not include any elasto-brittle (i.e., damage) or memory-effect models, if the focus of your paper is to support or not the inclusion of these memory effects.

Line 76: The reference to Marsan et al., 2004 is misplaced: they did not discriminate between different sea ice rheologies, they analyzed observations of deformation statistically.

Lines 78-83: Do the Bouchat et al. 2022 and Hutters et al, 2021 papers really conclude that the differences in the capability of these models to reproduce observations is due to their spatial discretization, i.e., FEM vs FD, or is this an interpretation? Then, what is the explanation for the factor of 5 between the resolutions required for FEM vs FD?

Line 87: A clarification here: ''elastic deformations prior to fracture''. In all EB-type models, the material is always elastic, prior to and *after* damage. In the MEB and BBM case, the material is visco-elastic and it becomes predominantly viscous, i.e. the relaxation time decreases after extensive damage.

Line 105: About neglecting the advection term because of spatial resolution: isn't it a question of temporal resolution?

Line 138: ''and viscous relaxation time $\eta$ and $\lambda$''. $\eta$ is the viscosity, $\lambda$ the viscous relaxation time, defined as the ratio of the viscosity and elastic modulus. They are not both relaxation times (and only two variables are independent).

Line 154-155: ''Its purpose is to convert the excess stress into damage (d)''. See general comments: its physical "purpose" is to redistribute stresses within the material, i.e., there is a *propagation* of stress perturbations.

Equation 16: I am a bit confused here: is there a parenthesis missing on the first term of the RHS of this equation? Otherwise, you could write the RHS as $= 1/t_d - f(zeta)/t_d - d/t_d - d/t_h$. Hence you have two constant (or at least damage-independent) terms, one damage-dependant damage rate term and one damage-dependant healing rate term. But these two last terms are of the same sign, whereas healing should in principle offset damaging. In your equation, the only positive term, incrementing damage, would be the first one, $1/t_d$, a constant. Can you clarify or perhaps expend on the meaning of each term in the equation? In any case, "d" in this equation is not damage, but a memory parameter

Line 171-173: See former comments about time scales in your model.

1. You choose a damage time scale that is "not bounded by the propagation speed of elastic waves", which in a brittle material that breaks, sets the upper bound for the speed of propagation of fractures. However, on what physical basis do you choose 1 day? How is this time linked to the mechanical behavior you assume in your model, i.e., viscous-plastic? You state that it is a ''typical time scale for fracture propagation''. Are there any observations or references supporting that? The information is also incomplete: if you give a time scale, you need to give an associated space scale over which propagation is happening, i.e., it takes a fracture one day to travel what distance?
2. You have a healing time, $t_d$, that varies between 2 days and 30 days. However, according to your equation 16 (last term on the right-hand-side), the healing rate in your model is not a constant but is damage-dependent (more intensively-damaged elements heal faster), making the healing rate space and time-dependent. This is an important point, that is not but should be mentioned in the text. Also, can you provide a physical explanation for the damage-dependance you are proposing for your healing rate?
3. Assuming that your healing rate does vary between 2 days and 30 days (eg., if d = 1): this is only 2 to 30 times slower than the damage propagation time scale. Recall that the healing time is the time it takes for ice over a completely damaged model element or grid cell (d = 1) to recover completely its undamaged (d = 0) state. One of the key points in the previous MEB (or BBM) framework is that there is a very large separation of time scales between damaging and healing (ex., 10^5-10^6, see [*Dansereau et al.*, 2016; *Weiss and Dansereau*, 2017]). If it is not met, damage does not "have time" to propagate before elements heal again, interactions between damaged parts of the ice are hindered, and the memory effect vanishes, obviously. You mention that in lines 457-459, but it would facilitate the interpretation of your results if this mention come earlier in the text.

Lines 175-179: ''Note that a VP model is a nearly ideal plastic material, i.e. it can be considered as an elastic-plastic material with an infinite elastic wave speed; therefore, the fracture propagation is instantaneous (i.e., it is resolved with the outer loop solver of an implicit solver or the sub-cycling of an EVP model). In the above equation, n is a free parameter setting the steady-state damage for a given deformation state.'' See general comments above about distinguishing between the equations and the numerics and about the contradiction between including a damage evolution equation together with a steady-state stress equation, and actually including the concept of damage within a VP framework.

Line 316: "soup-like": fluid-like?

Line 317: "except when th ≈ td when fewer extreme deformation events are present." See previous comment: This is rather trivial. At each time step, over completely-damaged elements

(d = 1), you then offset damage by the same amount of healing: there is no net effect of damage nor healing.

Lines 392-400: Interesting. See general comments above for possible insights about time scales and temporal behavior.

References:

Brenguier, F., M. Campillo, C. Hadziioannou, N. Shapiro, R. M. Nadeau, and E. Larose (2008), Postseismic relaxation along the San Andreas fault at Parkfield from continuous seismological observations, *Science*, *321*(5895), 1478-1481.

Dansereau, V., J. Weiss, P. Saramito, and P. Lattes (2016), A Maxwell–elasto–brittle rheology for sea ice modeling, *The Cryosphere*, *10*, 1339-1359.

Eshelby, J. D. (1957), The determination of the elastic field of an ellipsoidal inclusion, and related problems, *Proc. Roy. Soc. A*, *241*, 376-396.

Kachanov, L. M. (1958), Time of the rupture process under creep conditions, *Isv. Akad. Nauk. SSR Otd Tekh. Nauk.*, *8*, 26-31.

Kachanov, M. (1994), Elastic solids with many cracks and related problems, *Advances in applied mechanics*, *30*, 259-445.

Karimi, K., D. Amitrano, and J. Weiss (2019), From plastic flow to brittle fracture: Role of microscopic friction in amorphous solids, *Phys. Rev. E*, *100*(1), 012908.

Kondo, D., H. Welemane, and F. Cormery (2007), Basic concepts and models in continuum damage mechanics, *Revue européenne de génie civil*, *11*(7-8), 927-943.

Lemaitre, J., and J. L. Chaboche (1990), *Mechanics of solid materials*, Cambridge University Press, Cambridge.

Weibull, W. (1939), A statistical theory of the strength of materials, *Proc. Royal Swedish Academy of Eng. Sci.*, *151*, 1-45.

Weiss, J. (2013), *Drift, deformation and fracture of sea ice - A perspective across scales*, Springer, Dordrecht, The Netherlands.

Weiss, J., and V. Dansereau (2017), Linking scales in sea ice mechanics, *Phil. Trans. R. Soc. A*, *375*(2086), 20150352.

Weiss, J., L. Girard, F. Gimbert, D. Amitrano, and D. Vandembroucq (2014), (Finite) size effects on compressive strength, *PNAS*, *111*(17), 6231-6236.

---

## Author Comment (AC1)

**Editor**

**Originality (Novelty):** 2. In recent years there has been considerable interest in the scaling properties of sea ice fracture and deformation and the ability of models to reproduce this. While I find the premise of this paper, the introduction of a damage parameter into the VP rheology, to be somewhat poorly motivated from the perspective of the proximate physical processes, the VP model itself is not argued closely from proximate physics in any case. The inclusion of damage therefore seems a legitimate extension for the purpose of generating improved understanding of the links between rheological assumptions, resolution, and potentially other factors in generating multifractality.

**Scientific Quality (Rigour):** 2. My rapid review suggests that a methodical approach has been taken with due care and attention. However, I shall rely on detailed reviews to assess the scientific rigour of the manuscript.

**Significance (Impact):** 3. While there is legitimate academic interest in the scaling properties of fracture, it is not convincingly explained why these are of practical interest, e.g. to climate modelling or operational needs. The non sequitur at the end of the second paragraph ("As such..") in the introduction does not, in of itself, bear up to critical analysis, though the argument could be made more convincing and would likely open up interest more broadly.

We agree. We added, before the last sentence of the paragraph at line 28, the following: "These physical processes must be adequately represented in both global climate models and ice–ocean prediction systems either through added physics or parameterizations. The ability of a sea-ice model to reproduce Linear Kinematic Features includes length, angle between conjugate pairs of LKFs, lifetime, and statistical spatiotemporal properties such as scaling, multifractality, coupling, etc."

**Presentation Quality:** 2. The paper seemed generally well written, nicely dissecting the relevant issues, and is logically organised. The opening sentences seem more appropriate for a general audience than the readership of The Cryosphere but are more amusing than distracting.
* * *
The manuscript proposes a simple, but ad hoc and poorly physically justified, approach to modification of the ice strength in a popular sea ice rheology scheme. The purpose of this is to create a low-computational cost way to better simulate some of the scaling properties of deformation. It is interesting, though not especially surprising (as the authors note), that this enables capture of some of these scaling relations with such a simple treatment.

I am sympathy with the general approach taken and there is much in this paper that is of value. The ad hoc approach is obviously not ideal and raises important concerns over appropriate choice of timescales, but then the rheology that is being modified is also exceedingly ad hoc and has been in use for over 40 years. Clearly the authors are aiming for a no-cost, practical approach that can be easily implemented into existing modelling architectures.

The development of the elasto-brittle (EB) and modification thereof (MEB and BBM) raised several interesting questions that have not been addressed by the developers of the EB family, namely, what part of the simulation results are due to the numerical implementation (finite element and Lagrangian advection scheme vs finite difference, fixed grid and eulerian advection scheme in the traditional VP approach)? What part of the results are due to resolved model physics (elasticity in this case) and ad hoc parameterization (the damage)? The current paper is the third of a series of three papers trying to address these questions. In the first one, Bouchat et al (2022) corrected non-reproducible results or unsubstantiated statements reported in Girard et al (2011) and Weiss and Dansereau et al. (2016), namely: "In particular, the VP rheology, currently used in most sea-ice models, has been shown to be unable to capture the properties of ice strain rates." and "... [the VP model] fails at reproducing the observed properties of sea ice deformation ...". In the second paper by Plante et al, (2022), the MEB was coded in finite difference on a fixed grid with an Eulerian advection scheme as a platform to address the first question. The main goal of the current paper is to disentangle the effect of resolved physics (plasticity in this case) from parameterized physics (damage) in the context of the VP model where both can be turned on and off independently. The no-cost, practical approach that can

be coded easily in existing architectures is a secondary goal/outcome of the current work. This was stated in the abstract at line 1, in the introduction at lines 88–92, and has been reworded to convey this message more clearly. In the abstract at line 1 we have: "We implement a damage parametrization in the standard viscous-plastic sea ice model to disentangle its effect from resolved model physics (visco-plastic with and without damage) on its ability to reproduce observed scaling laws of deformation."

Nonetheless, one of the reviewers raised very important points that must be taken into account. Particularly damage refers to reduction of the elasticity, not plastic ice strength. As an Editor, I view it as my job to prevent the propagation of terms that are likely to cause confusion and are at odds with existing and mature bodies of work. As a result, I will not accept this paper if the word damage is used in its current form.

We agree. The distinction between the commonly accepted definition of damage and the definition used in the manuscript was not clearly spelled out. The goal of damage in models is to simulate large deformation for a given stress level when bounds between molecules are broken. In the context of linear elasticity theory, this can only be accomplished by reducing the shear or young modulus of the material. In the context of sea ice and plasticity theory, this can only be accomplished by including strain weakening (independently from divergence and a subsequent reduction of ice thickness and concentration). The term damage however is used in both communities and referred to as elastic and plastic damage, respectively (see for instance Lubliner et al., 1989; Friedlein et al., 2023; Jason et al., 2006; Voyiadjis and Taqieddin, 2008; etc. for (elastic-)plastic damage literature). We now introduce at line 162 the term as "plastic damage (hereafter referred to as damage for simplicity)", followed by the sentence: "In the context of sea ice and plasticity theory, plastic damage (hereafter referred to as damage for simplicity) can be accomplished by including strain weakening in the model that is independent of subsequent divergence and reduction of ice thickness and/or concentration (see for example Lubliner et al., 1989 for a simple model of plastic degradation). In recent years, more complex models have been developed that include (elastic-)plastic damage, notably in concrete (see for instance Luccioni et al., 1995; Jason et al., 2006; Voyiadjis and Taqieddin, 2008; Parisio and Laloui, 2017; Hafezolghorani et al., 2017; Friedlein et al., 2023; etc.) in which plasticity is taken into account in the damage variables, and the yield curve changes accordingly. Since elasticity is not taken into account in the VP model, we use a parametrization of damage that changes the yield curve of sea ice depending on its damage level extracted from its viscosity rather than its elasticity. Therefore, the damage that we present here, while intuitively based on previous literature, remains a parametrization of a more complex mechanism."

If the authors wish to proceed with revisions then careful account should be taken of all the comments by the reviewers, with particularly full responses given if a recommendation is not followed. But I reiterate that an alternative language must be found to express the reduction in ice strength and the word damage removed from the paper and title. The obvious term that springs to my mind is plastic weakening, which is what I believe it actually is.

---

## Author Comment (AC2)

**Reviewer 2**

This paper is a useful contribution because the authors have demonstrated the multifractal behavior the VP and a VP model modified to include damage can have (or in fact not have). While the model does not match observations there is discussion of how the model behaves that could help future investigators improve models of sea ice deformation. What is lacking is often a physical interpretation of the model behavior. While relationships are described mathematically, such as the differences in scaling exponents or departure from expected multifractal behavior, no context as to why this might be is given. Perhaps this is not possible to determine, however it would be useful insight.

Overall I feel this contribution is valuable and should be published. Though I would caution readers to consider if multifractals are the best metric to validate simulated deformation or distinguish models. There are places where the authors can strenghten their argument as to why it is important to reproduce observed scaling behavior for sea ice deformation or linear kinematic features. The contribution of this paper in describing the model behavior with the mathematical language of multifractals is helpful in the conversation.

**General comments**

The deformation of sea ice is shown to be coupled in space and time, such that the scaling relationship in each varies depending on the sampling in the other. This means that when comparing a model with observations you have to account for this time or space sampling difference. How do you ensure that you are comparing scaling relationships for the same time or space sampling in the model and observations?

These are important considerations for sea ice but are treated in great detail in other works. We include a brief discussion on this topic after line 30: "As discussed in Weiss, (2013), the scaling properties of sea ice, contrary to those of turbulence in fluid, have been shown to be coupled in space and time such that the value of one is influenced by the sampling resolution of the other, and also by its spacetime localization." For the second point, we direct the reviewer to section 4.3 for the method that we use to compute the scaling statistics. In this, we clearly state how we compute the scaling, for $\beta(T = 3 \text{ days})$ and $\alpha(L = 10 \text{ km})$, as stated in line 269 and 279 respectively.

It is a useful comment that even ad-hoc parameterization of heterogeneity in ice strength could improve representation of sea ice deformation. This jives with my personal experience where if I solve the VP model to full plastic equilibrium it is not possible to simulate LKFs unless you seed variability in ice strength randomly. Which is about as simple a parameterization as one can make! In the introduction a damage parameterization is introduced, and in the discussion this is compared to other parameterizations used in other models. It might help the reader to give a little information about how the parameterizations differ up front and why you choose to develop your own.

A VP model can simulate LKFs without seeding variability in ice strength (keeping the ice thickness and concentration constant). See for instance Ringeisen et al. 2019) who simulate conjugate pairs of LKFs with a fracture angle that is in accord with theory. When seeding "defects" in the ice, they show other LKFs linking defects together with a fracture angle that departs from theory. There appears to be something wrong in the way the reviewer has run their model. Girard et al (2011) were also not able to simulate spatial scaling with their VP model, in contrast with results from all other groups currently running a VP model (Bouchat et al. 2023). Our VP model is publicly available on github https://github.com/McGill-sea-ice/SIM, in case it can be useful to check on the reproducibility of results in the future.

We now introduce at line 162 the term as "plastic damage (hereafter referred to as damage for simplicity)", followed by the sentence: "In the context of sea ice and plasticity theory, plastic damage (hereafter referred to as damage for simplicity) can be accomplished by including strain weakening in the model that is independent of subsequent divergence and reduction of ice thickness and/or concentration (see for example Lubliner et al., 1989 for a simple model of plastic degradation). In recent years, more complex models have been developed that include (elastic-)plastic damage, notably in concrete (see for instance Luccioni et al., 1995; Jason et al., 2006; Voyiadjis and Taqieddin, 2008; Parisio and Laloui, 2017; Hafezolghorani et al., 2017; Friedlein et al., 2023; etc.) in which plasticity is taken into account in the damage variables, and the yield

curve changes accordingly. Since elasticity is not taken into account in the VP model, we use a parametrization of damage that changes the yield curve of sea ice depending on its damage level extracted from its viscosity rather than its elasticity. Therefore, the damage that we present here, while intuitively based on previous literature, remains a parametrization of a more complex mechanism."

**Specific comments**

**Abstract line 8:** Grammatical error "an"

Corrected as suggested by the reviewer.

**Line 14:** Is "ilks" a good word to use here? This is a stylistic comment that you can take or leave: I found the first paragraph of the introduction did not really guide me to what the content of the paper would be about. In general I would suggest the introduction could be more focused.

We replaced "ilks of ice" with "countless variations of ice and snow". It is correct, the paragraph does not guide the reader to the content of the paper. The current uniform/direct/bare-bone scientific writing style was introduced in the 70s for the sake of uniformity between journals and discipline, but in doing so, we have lost some of the authors' perspectives and states of mind (read, for instance, earlier AIDJEX literature which was much more personal). We believe that something was lost in this transition. The editor has raised a similar issue. We decided to keep the paragraph, but we keep the suggestion in mind for the future.

**Line 24:** Typical floe sizes range from meters to 10s of kilometers. So saying floe size is 10km is factually incorrect. It is correct to mention that the data you are working with has this lower resolution, but incorrect to call it floe size.

This is correct. The sentence was changed from "ranging from floe size (10 km) to the size of the Arctic Basin" to "ranging from the smallest floes (meters) to the size of the Arctic Basin".

**Line 30:** "complex laws" This is overly general. I encourage you to be more specific. This is also where you could point out that unlike turbulence in water or air, the scaling relationships for sea ice deformation are coupled in space and time. There is also large differences in scaling exponents found for different years and seasons, so it would be good to comment on if you accounting for this or just using values found for a particular time period or region.

See the first paragraph of the general comments, we added a small discussion on this topic after line 30.

**Line 34:** Why should a model of Arctic sea ice simulate LKFs?

The reasons why a proper simulation of LKFs are important are stated in the previous paragraph. In this paragraph, we are coming back to the same idea and elaborate on the exact reasons. We now state all the reasons why a proper LKF simulation is important in the same paragraph when it is first mentioned.

**Line 51:** Missing full stop.

The reviewer is referring to: "Models that incorporate some of these recommendations include the Elasto-Brittle and modification thereof (EB, MEB, and BBM) [references], in which elastic deformations are followed by brittle failure, while larger deformations along fault lines following damage build-up are viscous." There is a period at the end of the sentence, please clarify.

**Line 163:** Sea ice can diverge and weaken (through reduced area) in the VP model even when LKFs are not present.

This is correct. We clarified that we are referring to weakening along LKFs. The new sentence reads: "weakens along an LKF only when sea ice divergence is present".

**Line 205–210:** I am curious, when you are creating a run with 10 random years how do you ensure there are not unphysical jumps in the wind forcing between years? Does this matter, given the spin up of ice drift is relatively fast compared to the wind speed change.

There are discontinuities at the end of each year. But the response time of sea ice velocities and thickness to changes in forcing is fast and the simulated sea ice deformation is not sensitive to exact sea ice thickness conditions (see for instance Bouchat and Tremblay, 2017 who simulated sea ice deformation for different years with different ice thickness distribution). The advantage of using random years is to avoid spinning up the model for a series of 10 years in a given phase of, say, the Arctic Oscillation or other low-frequency variability which would lead to a larger difference in the initial ice thickness field. This was clarified on line 238 of the revised manuscript.

**Line 316:** "soup-like" is a weird choice of word here. Also is there a missing label by the "2".

Corrected as suggested by the reviewer. We now use Figure 2d.

**Line 320:** "simulation" $\implies$ "simulations"

We are referring here to only one simulation with no damage and increased shear strength ($e = 0.7$). The sentence was changed to: "as in the simulation without damage with $e = 0.7$".

**Line 331:** I believe the decay is log-linear not linear.

Correct. we are referring to a log-linear decay. This is clarified: "This shift results in a log-linear decay in the tail of the PDFs [...]"

**Line 378:** "morally" is a weird word choice here.

We agree. We replaced "being morally the same as that of RGPS" with "being the highest (0.14) and closest to that of RGPS".

**Figure 3:** It is not clear where the observations in the black line are from. Where do you describe the observations in the paper and how you calculate the deformation. Are the distributions for model and observations from the same time and region? It is only apparent later in the paper that you are only considering data from January 2002 (line 538). Which makes me think you need to improve details in your methodology.

Thanks for pointing this out. We now state in the Methods (section 4) that the simulated sea ice deformations are validated against RGPS data for January 2002 (following Bouchat and Tremblay 2017). This was clarified on line 230 of the revised manuscript:

"We calculate all metrics using simulated 3-day average sea ice velocities inside the SAR sea ice RGPS data in the region where an 80% temporal data coverage is present for the winters 1997–2008 [...]. The sea ice deformations are compared with observed sea ice deformations derived from RGPS three-day average ice velocities for January 2002. The results presented are robust to the exact choice of year.."

**Data:** I could not check code and data availably because links were not provided.

The code and data will be included when the paper is accepted for publication.

---

## Author Comment (AC3)

**Reviewer 1**

**General comments**

In this manuscript, the authors propose to introduce in a viscous-plastic (VP) sea ice model a "memory" parameter that they call (inappropriately, see below) "damage", making the ice weaker (i.e. its strength P, setting the size of the yield curve, decreases) each time it yields. This generates a positive feedback leading to a weaker and weaker ice, an effect counterbalanced in their model by "healing" (eq. 15). The authors implement this scheme into their VP model and run 1-month simulations over the Arctic forced by geostrophic wind fields. They compare the results of their model with "classical" VP simulations (i.e. without this memory effect) in terms of strain localization and scaling, and argue that this new parameterization improves the modelling of strain localization (which makes sense considering the positive feedback mentioned above) and spatial scaling, but not the temporal scaling.

I have little comments about the numerical scheme, the forcing, etc... I have some about the evaluation and the comparison with RGPS data (see below). My main concern, however, is about the concepts. The authors define their memory parameter as "damage", and argue that it is equivalent to the damage parametrization in the EB/MEB family of models.

Nowhere in the paper is stated that the damage parameterization "is equivalent to the damage parameterization in the EB/MEB family of models". If anything, the exact opposite is said in one location at line 464: "This is of course expected since damage in the VPd model does not represent necessarily the same thing as damage in the MEB model". The paper simply states that damage can be parameterized in VP models, and we propose one way to accomplish this goal. The only thing that is akin to the EB-family damage is its temporal evolution equation, which is cast in a similar form. This is clarified on line 166 of the revised manuscript as: "We include damage in the VP model using a simple advection equation with source/sink terms *(akin to what is used in the MEB formulation)* of the form ...". So that it is clear that what is like the MEB is the evolution equation, not the damage per se.

Since the seminal paper of [L M Kachanov, 1958], damage mechanics became an actual branch of solid mechanics with (at least) one international journal devoted to it, many classical works (e.g. [M Kachanov, 1994; Kondo et al., 2007] among many others) and several books (e.g. see [Lemaitre and Chaboche, 1990]). In all these works, damage refers, in continuum mechanics, to a degradation of elastic properties as the result of emerging internal defects (microcracks, voids,..), an effect that is homogenized at the considered scale. Note that damage (with this true definition) can also be extended to granular media [Karimi et al., 2019], without the need to break bonds or particles, just from topological rearrangements between particles, that effectively lead to elastic softening.

The reviewer is citing literature on elastic damage. There is an equally large body of work on plastic damage describing how (plastic) damage can be included in plastic models. These models are usually elastic-plastic models, and consider both elastic and plastic damage; in the case of the VP model, only plastic damage is considered. Therefore, in reality, one usually defines damage as being in part due to elasticity (change in the Young's modulus) and in part due to plasticity (change in the yield curve) and, therefore, when the reviewer says "with this true definition", this is in fact ignoring a significant portion of the literature on damage mechanics. As Max Born said: "The belief that there is only one truth and that oneself is in possession of it seems to me the root of all the evil that is in the world."

The memory parameter introduced by the authors has therefore nothing to do with damage, which, by construction, cannot be distinguished implemented in a model that ignores elasticity. This is not just a matter of semantics: See e.g. L61-65 "Damage parametrizations — first developed in rock mechanics — are ad-hoc in that they are not derived from observations and/or from first physics principle". This is a very strong, yet false statement. Damage has actually been formulated from first physical principles, observed, measured in many communities (mechanics, geomechanics, geophysics (e.g. from a decrease of elastic wave velocities around major faults after an earthquake [Brenguier et al., 2008]), etc,see the literature suggested above). Consequently, in its present form, this manuscript would falsely perpetuate the idea within the sea ice and perhaps other communities that damage is an "ad-hoc" parameterization, and doing so, would ignore only about 65 years of work.

The reviewer is arguing that damage mechanics comes from first physical principles. This is incorrect as some form of smoothing has to be performed to extract continuum-scale variables out of the damaged system. The modeling of damage through the variation of elastic modulus is one of many ways of doing this. One could also model it by reducing the effective area of the sample, or by looking at void volume fraction for instance. In the paper cited by the reviewer [Brenguier et al., 2008], no mechanism is provided as to why an initial abrupt increase in elastic wave speed is followed by a slow decrease, *in seismic activity*. They hypothesize that the observed behavior is governed by postseismic stress relaxation within deeper parts of the fault zone. It could be due to an increase in elastic modulus, followed by a decrease (in the upper crustal layers), but the observed behavior could also be due to a change in density (elastic wave speed is a function of density) associated with the rearrangements of relatively high-density pockets (phase transition) in deeper parts of the mantle (Ringwood, 1972). Moreover, it is not because something is true in rocks mechanics that is true for sea ice. The damage in the context of sea ice modeling is formulated as a simple relaxation equation with one free parameter $\alpha$ applied to the viscosity and relaxation time. If it came from first physics principles, the parameter $\alpha$ would have been given a name other than a "constant" (e.g. Dansereau et al., 2016).

On the reverse, the memory effect introduced here is indeed an ad-hoc parameterization. Note that in the MEB framework, the dependence of the viscosity on damage is partly empirical as well, within some physical bounds.

The reviewer states that the damage parameterization is partially ad hoc or empirical. This is precisely the point we are making. The damage parametrized in the MEB comes from the definition of elastic damage, which locally changes the Young's modulus. In a complete elastic-plastic model, one also has to consider plastic damage, which modifies the yield curve of the material. In our case, we parametrize the plastic damage.

In the EB/MEB modelling framework, damage was introduced with two objectives:

- introduce indeed a memory of past loading history. The difference with the present work is that the elastic properties are evolving through time, i.e. following damage mechanics, not the strength.

Exactly, this is how plastic damage works.

- even more importantly, the objective was to propagate elastic stresses within the elastic medium following a damage/fracturing local event, this way allowing fracturing itself to propagate in an appropriate way. This comes back to the historical work of [Eshelby, 1957]: an inclusion within an elastic body, with elastic constants within the inclusion different from the surrounding medium, generates an elastic stress redistribution kernel. This is exactly what is modelled within the EB/MEB framework, and obviously not within the VP framework, with or without a memory parametrization. This fundamental difference should be clearly stated in the manuscript to avoid confusion.

As stated by the reviewer in the comment below, the VP assumes steady state between stresses and strain, and this redistribution of stress is taking place within the outer loop solver (at one given time step). The sentence (on line 171): "In contrast with the MEB model, damage is not bound by the propagation speed of elastic waves." was changed to "In contrast with the MEB model, damage is not bound by the propagation speed of elastic waves, *and damage will propagate within the outer loop solver, as opposed to being resolved with a finite elastic wave speed.*"

Still on concepts: to imagine that a material previously damaged will yield/fail more easily than an intact material seems to make sense at first glance. However, the relation between damage and strength is far to be trivial, certainly non-linear, most likely strongly dependent on the material, the loading geometry etc... Actually, progressive damage models have been used to explore this complex physics, and showed that strength generally results from complex (including long-ranged elastic) interactions between defects, resulting in non-trivial size effects [Weiss et al., 2014]. Even if ignoring the interactions between defects, the ultimate strength of a solid will depend on the (extreme) statistics of the population of defects, actually on the "largest" flaw in the [Weibull, 1939] classical approach, and not on homogenized "damage". Regarding the parameterization of memory implemented in this paper, it also poses some contradictions with the VP constitutive equation. One way to understand this contradiction and the incompatibility of the

parameterization with actual damage is that the authors implement a time-evolving "damage" evolution equation (with a given propagation time scale). Damage, by definition, measures the effective stress and its propagation is therefore an image of stress redistribution. However, the viscous(-plastic) constitutive equation intrinsically assumes a steady state between stress and deformation rate. In other words, it does not account for stress redistribution: there is no d$\sigma$/dt (propagation, or elastic) term, only a $\sigma$ (diffusion, or viscous) term.

The reviewer is again referring to elastic damage. Plastic damage is simulated as a change in the plastic flow vector. The definition of elastic damage provided by the reviewer does not apply to our parametrization. In this context, it is true that what we are doing is not *elastic-based* damage. We now introduce at line 162 the term as "plastic damage (hereafter referred to as damage for simplicity)", followed by the sentence: "In the context of sea ice and plasticity theory, plastic damage (hereafter referred to as damage for simplicity) can be accomplished by including strain weakening in the model that is independent of subsequent divergence and reduction of ice thickness and/or concentration (see for example Lubliner et al., 1989 for a simple model of plastic degradation). In recent years, more complex models have been developed that include (elastic-)plastic damage, notably in concrete (see for instance Luccioni et al., 1995; Jason et al., 2006; Voyiadjis and Taqieddin, 2008; Parisio and Laloui, 2017; Hafezolghorani et al., 2017; Friedlein et al., 2023; etc.) in which plasticity is taken into account in the damage variables, and the yield curve changes accordingly. Since elasticity is not taken into account in the VP model, we use a parametrization of damage that changes the yield curve of sea ice depending on its damage level extracted from its viscosity rather than its elasticity. Therefore, the damage that we present here, while intuitively based on previous literature, remains a parametrization of a more complex mechanism."

On the more specific question of time scales, the authors propose a time scale for the propagation of "damage" without justifying it on a physical basis, or specifying the associated space scale considered. This is missing from the paper and has repercussion on the interpretation of the appropriate, nominal healing time scale (see specific comments below). As in previous works on visco-elastic-brittle-type models, the conclusions of the authors however support the requirement of a clear separation between the damaging and healing time scales. The fundamental underlying contradiction arising from the absence of such separation (healing almost as fast as fracturing) should be introduced much earlier in the text.

The reviewer is referring to the following sentence: "The choice of a small damage timescale comes from the synoptic timescale at which fractures develop, while a large healing timescale comes from the thermodynamic growth of one meter of ice." The revised sentence now references the spatial scale and reads (see line 173 of the original manuscript): "The choice of a small damage timescale comes from the synoptic timescale at which fractures develop *over tens of kilometers*, while a large healing timescale comes from the thermodynamic growth of one meter of ice."

Concerning parts 4 and 5 and the comparison of model results to RGPS data in terms of strain localization as well as space and time scaling:

- I do not understand the interest to compare models outputs to observation in terms of PDFs (5.2) and then CDFs (5.3). The very same information is contained in both.

We keep both the PDFs for comparisons with existing literature and the CCDF (see below) because it supports the discussion better.

- On figure 4, the data seem to have been plotted (or accumulated) in a wrong way, i.e. $P(< X)$ is shown, not $P(> X)$. This should not be done that way, as small strain-rate values (at least, in the observed dataset) suffer from signal/noise biases, so we are more interested by the upper tail of the distribution. Consequently, I do not understand the associated evaluation (section 5.3).

Good point, thanks for noticing this. We are plotting the CCDF ($P(X > x)$. The figures are now flipped vertically (removing the confusion). The caption now correctly refers to CCDF, instead of CDF, and the y-axis was changed for a log scale to highlight the differences between the various configurations in the tail of their distribution (see above).

- Overall, the authors report some "scaling", or at least a scale-dependence of strain-rates. This, in my sense, is not completely enough to argue that the model incorporates the right physics (or not). Indeed, the models

(VP or VPd) are here forced using geostrophic wind fields which are themselves characterized by some complexity. And so spatial gradients in the forcing fields will inevitably induce some strain localization. The memory effect introduced here seems to reinforce this, which appears reasonable indeed. A real test would be to force such model with a homogeneous field. In that case, spatial scaling naturally emerges using progressive damage models, as the result of elastic long-ranged interactions and threshold mechanics (so, the introduction of damage).

We cannot find this study showing that the spatial scaling emerges naturally in progressive damage models in the peer-reviewed literature. Please give the reference.

The shear strain in the geostrophic wind field scales like $[U]/L = 10\,\mathrm{m/s}\ /\ 500\mathrm{km} = 2 \times 10^{-5}\,\mathrm{sec}^{-1}$, where U is a characteristic wind speed and L is the synoptic length scale. Applying a 2% transfer function between wind speed and free drift ice velocity (Brunette et al. 2023), this gives a shear strain rate in the sea ice of $4 \times 10^{-7}$ which is to the left of the tail end of the PDF associated with LKFs.

Nevertheless, some important differences between sea ice scaling properties and fluid turbulence exist, such as a space/time scaling symmetry, which is not discussed here. See e.g. [Weiss, 2013] for a discussion on this topic.

Indeed, these are important considerations for sea ice but are treated in great detail in other works, as the reviewer pointed out. We include a brief discussion on this topic after line 30: "As discussed in Weiss, (2013), the scaling properties of sea ice, contrary to those of turbulence in fluid, have been shown to be coupled in space and time such that the value of one is influenced by the sampling resolution of the other, and also by its spacetime localization."

In conclusion, to the question "Can we introduce a memory effect in VP sea-ice models from an empirical parametrization of the strength evolution?", I would say, why not ? , and it could indeed improve, empirically, the representation of sea ice deformation fields to some extent. However, the manuscript in its present form cannot be recommended for publication, for all the conceptual reasons given above. First, the term "damage" should be removed everywhere from the text, including in the title, as this manuscript is not dealing with this concept (see above). Then, a clear distinction between the concepts used here and in EB/MEB should be stated clearly, following the comments above.

The term damage was used in previous sea ice modeling work, following rock mechanics conventions, to mean "elastic damage", i.e. a reduction of the elastic modulus as opposed to a reduction in strength. Damage was also used in the simulation of plastic deformation in concrete and other materials, and referred to as "plastic damage". We now, clearly define the terms elastic and plastic damage up front and state that the term "damage" used in the text refers to "plastic damage", akin to published articles on the EB/MEB/BBM models that refer to "elastic damage" as damage.

**Specific comments**

**Abstract:** "We implement a damage parametrization in the standard viscous-plastic sea ice model to disentangle its effect from model physics (visco-elastic or elasto-brittle vs. viscoplastic)". This sentence is misleading. First, how is damage not part of the model physics? Second, you do not compare a visco-plastic model with a visco-elastic model here that includes or not damage. Please rephrase.

A parameterization is not part of the model physics (rheology), by definition. It is a process that is not resolved by the model that is implemented in terms of variables that are resolved by the model. A parameterization can be inspired from observations (e.g., heat conduction) or physics (e.g., damage), but it is not model physics. We are comparing two VP models one with and one without damage. We have removed "visco-elastic or elasto-brittle" in the parenthesis and the revised text now reads: "(visco-plastic with and without damage)".

**Line 21:** "Internal stresses rapidly redistribute these forces from ice–ice interactions over long distances." This is exactly what an elasto-brittle model (based on elasticity and progressive damage) is built to represent, and what a VP model does not consider, either with or without a memory effect included.

Here, we are describing the behavior of real-world sea ice, independently of the model physics. As to the above comment, stresses are redistributed rapidly by elastic waves in an elasto-brittle rheology and instantly in a VP rheology, i.e., within the outer loop solver.

**Line 35:** "sea ice motion". And deformation.

Corrected as suggested by the reviewer.

**Lines 36-40:** "sea ice is considered as a highly-viscous fluid for small deformations. In this case, sea ice deforms as a creeping material." For small "stresses", instead. Please clarify.

Sea ice behaves as a viscous material for any stress that lies within the yield curve. These can be equally large as the plastic stress on the yield (to within a small epsilon). The viscous regime is really applicable to small deformation not just small stresses. This sentence is standard in all publications on the VP model.

**Line 48:** "the inclusion of deformation on discontinuities": not clear. What do you mean? Deformation or discontinuities should not be introduced in an ad-hoc manner, but result from the modelled rheology.

We were referring to Coon et al. 2007 statement: "Displacement data show that discontinuities in velocity caused by lead opening, closing, and shear must be accounted for in the representation of deformation." But we agree it is unclear. The revised sentence now reads: "For instance, ice would be better described using a rheology that accounts for discontinuities in the velocity field, and an anisotropic yield curve that allows for tensile stresses \citep{coon2007}".

**Line 49:** "anisotropic yield curve *that allows* tensile stresses".

Corrected as suggested by the reviewer.

**Line 51:** "Finite element models...". This should be taken as another main comment: Another source of confusion in the paper is the association of certain rheologies (EB-type, Elasticdecohesive) but not others (VP) to the numerical schemes that have been employed in models implementing these rheologies. This gives the wrong impression that there is some causal relation between the numerics and the model equations. These are actually two distinctive things: you first formulate a system of equations and if consistent, it is independent from the choice of spatio-temporal discretization scheme you then apply. In the discussion, you even extend your supposition of good/bad agreement with observations to the Eulerian vs Lagrangian framework in which the equations are cast. However, as for numerical schemes, you never offer and explanation as to why. These suppositions add up, so much so that the reader is lost: what is best between damage or not damage, memory or not, rheology vs numerical scheme. The comparison you make here allows distinguishing only between a VP model with and without a memory effect and that should be made clear.

The first implementations of the EB model family were done using a finite element approach with a Lagrangian advection scheme. The authors concluded that the better realism of the MEB was due to the elasto-brittle rheology but without justification. A justification of this statement would have included a comparison of the MEB rheology using a finite difference and a finite element model with a Lagrangian scheme and identify what part of the "better realism" is due to numerics (FEM and FD) and what part is due to physics. Plante et al. 2021 coded the MEB model in finite difference to answer that question. The sentence was changed to: "Models that incorporate some of these recommendations include the Elasto-Brittle and modification thereof (EB, MEB, and BBM) [references] *originally cast in* Finite Element numerical schemes (FEM) [...]".

**Line 53:** .."that damage associated with (prior) fractures also affects ice strength". No ! You are here making a confusion between elastic modulus and strength (stress at failure). If both are expressed in Pa, they are completely different concepts – see also general comments above.

Corrected as suggested by the reviewer to: "These [EB] models include a damage parametrization that accounts for the fact that damage associated with (prior) fractures also affects the elastic modulus of the ice [...]".

**Line 61-65.** Please remove this wrong statement (see above).

The reviewer is referring to: "Damage parametrizations — first developed in rock mechanics — are ad-hoc in that they are not derived from observations and/or from first physics principle. For instance, a damage parameter can be quantitatively expressed as a scalar relationship between the elastic modulus of a material before and after fracture \citep{amitrano1999diffuse}. In this model, the ice strength does not decrease when damage is present; instead, it is the Young's modulus that decreases, resulting in larger deformation for the same stress state." The first sentence now reads (modified text in italic), "Damage parametrizations — first developed in rock mechanics — are ad-hoc *in that they are inspired by observations and not from first physics principles.*"

The reviewer makes contradictory statements. In the first reviewer's paragraph on page 4: "On the reverse, ...", the reviewer states that "the dependence of the viscosity on damage is partly empirical". Then the reviewer wants us to remove a "false statement" (lines 61-65), even though these lines are saying what the reviewer said in his previous statement.

**Line 65:** "the damage is expressed as a function of the stress overshoot". Not clear: The (temporal) evolution of damage, or damage increment, is defined as a function of the stress overshoot.

Corrected as suggested by the reviewer.

**Lines: 69-70:** Again, you introduce some confusion between the definition of damage itself and its incrementation or propagation, or between damage and fracture mechanics. Please clarify.

The reviewer is correct. We clarify in the following way: "Other more complex descriptions of damage parameterizations (both in its definition and evolution) — such as fracture initiation around elliptical flaws [ref in the paper] — have been used in rock mechanics and could in principle be implemented in sea ice models."

**Lines 72-74:** "Earlier model–observation comparison studies, aimed at defining the most appropriate rheology for sea ice, found that any rheological model that includes compressive and shear strength reproduces observed sea ice drift, thickness, and concentration equally well (e.g. Flato and Hibler, 1992; Kreyscher et al., 2000; Ungermann et al., 2017)". It seems a little odd to cite here these early studies of rheological model comparisons, which do not include any elasto-brittle (i.e., damage) or memory-effect models, if the focus of your paper is to support or not the inclusion of these memory effects.

This sentence is there to introduce the reader to the necessity of looking at deformation statistics to differentiate rheologies.

**Line 76:** The reference to Marsan et al., 2004 is misplaced: they did not discriminate between different sea ice rheologies, they analyzed observations of deformation statistically.

Corrected as suggested by the reviewer.

**Lines 78-83:** Do the Bouchat et al. 2022 and Hutters et al, 2021 papers really conclude that the differences in the capability of these models to reproduce observations is due to their spatial discretization, i.e., FEM vs FD, or is this an interpretation? Then, what is the explanation for the factor of 5 between the resolutions required for FEM vs FD?

They suggest that the spatial discretization explains the differences and that more work is required to address this. A finite difference model needs 5-7 grid cells to spatially resolve discontinuities. The effective resolution of a finite element discretization with a Lagrangian advection scheme is therefore comparable to that of a finite difference model with a spatial resolution roughly an order of magnitude higher.

**Line 87:** A clarification here: "elastic deformations prior to fracture". In all EB-type models, the material is always elastic, prior to and after damage. In the MEB and BBM case, the material is visco-elastic and it becomes predominantly viscous, i.e. the relaxation time decreases after extensive damage.

Corrected as suggested by the reviewer: "[...] consideration of elastic deformations prior to and subsequent to fracture allowing the material to retain a memory of past deformations."

**Line 105:** About neglecting the advection term because of spatial resolution: isn't it a question of temporal resolution?

It is a function of both length scale and time step (temporal resolution). Advection scales like $u^2/L \approx 0.05\,\mathrm{m/s^2}/10^4\,\mathrm{m} \approx 10^{-7}$, which is 5 order of magnitude smaller than other terms in the momentum equation (see Zhang and Hibler, 1997)

**Line 138:** "and viscous relaxation time $\eta$ and $\lambda$". $\eta$ is the viscosity, $\lambda$ the viscous relaxation time, defined as the ratio of the viscosity and elastic modulus. They are not both relaxation times (and only two variables are independent).

Corrected as suggested by the reviewer. It now reads: "[...] (e.g., elastic stiffness E, viscosity $\eta$, and relaxation time $\lambda$) [...]".

**Line 154-155:** "Its purpose is to convert the excess stress into damage (d)". See general comments: its physical "purpose" is to redistribute stresses within the material, i.e., there is a propagation of stress perturbations.

In this sentence, we are simply reviewing the MEB framework and its numerical implementation; i.e. the temporal evolution of the stress state is solved using a finite time step, and when it lies outside of the yield curve, the stresses are brought back to the yield along a line that passes through the origin using a parameter $\Psi$ ($= \sigma_f/\sigma'$), where $\sigma_f$ is the corrected stress state lying on the yield curve and $\sigma'$ is the uncorrected stress state lying outside of the yield curve. $\Psi$ in turn is a source in the damage equation (see for example, Dansereau et al., 2016: "In the Maxwell-EB model, the change in level of damage corresponding to a local damage event is determined as a function of the distance of the damaged model element to the yield criterion."; or Rampal et al., 2016).

**Equation 16:** I am a bit confused here: is there a parenthesis missing on the first term of the RHS of this equation? Otherwise, you could write the RHS as = 1/td – f(zeta)/td – d/td – d/th. Hence you have two constant (or at least damage-independent) terms, one damage-dependant damage rate term and one damage-dependant healing rate term. But these two last terms are of the same sign, whereas healing should in principle offset damaging. In your equation, the only positive term, incrementing damage, would be the first one, 1/td, a constant. Can you clarify or perhaps expend on the meaning of each term in the equation? In any case, "d" in this equation is not damage, but a memory parameter

There are no parentheses missing. In the plastic regime, when $\zeta \approx 0$, $d(d)/dt = (1 - d)/t_d$, and damage increases asymptotically to a maximum value of 1. In the viscous regime, when $\zeta \approx \zeta_{max}$, $d(d)/dt = -d/t_h$, and damage asymptotes to zero with a time scale $t_h$. The two sentences above were added in the paragraph after Equation 16 of the revised manuscript.

**Line 171-173:** See former comments about time scales in your model.

1. You choose a damage time scale that is "not bounded by the propagation speed of elastic waves", which in a brittle material that breaks, sets the upper bound for the speed of propagation of fractures. However, on what physical basis do you choose 1 day? How is this time linked to the mechanical behavior you assume in your model, i.e., viscousplastic? You state that it is a "typical time scale for fracture propagation". Are there any observations or references supporting that? The information is also incomplete: if you give a time scale, you need to give an associated space scale over which propagation is happening, i.e., it takes a fracture one day to travel what distance?

We specify the spatial scale associated with damage. Line 173 of the original manuscript now reads: "The choice of a small damage timescale comes from the synoptic timescale at which fractures develop *over tens of kilometers*, while a large healing timescale comes from the thermodynamic growth of one meter of ice."

2. You have a healing time, td, that varies between 2 days and 30 days. However, according to your equation 16 (last term on the right-hand-side), the healing rate in your model is not a constant but is damage-dependent (more intensively-damaged elements heal faster), making the healing rate space and time-dependent. This is an important point, that is not but should be mentioned in the text. Also, can you provide a physical explanation for the damage-dependance you are proposing for your healing rate?

The reviewer is correct. We add at line 171: "Note also that the healing term in Equation 16 depends on the level of damage." A heavily damaged piece of ice would heal faster due to (but not limited to) having more

surface area exposed to water, and therefore, refreezing more easily.

3. Assuming that your healing rate does vary between 2 days and 30 days (eg., if d = 1): this is only 2 to 30 times slower than the damage propagation time scale. Recall that the healing time is the time it takes for ice over a completely damaged model element or grid cell (d = 1) to recover completely its undamaged (d = 0) state. One of the key points in the previous MEB (or BBM) framework is that there is a very large separation of time scales between damaging and healing (ex., $10^5$–$10^6$, see [Dansereau et al., 2016; Weiss and Dansereau, 2017]). If it is not met, damage does not "have time" to propagate before elements heal again, interactions between damaged parts of the ice are hindered, and the memory effect vanishes, obviously. You mention that in lines 457-459, but it would facilitate the interpretation of your results if this mention come earlier in the text.

We now include a discussion of this earlier in the text (after line 173): "The healing timescale should be much larger than the damaging timescale [as pointed out in, Dansereau et al., 2016; Weiss and Dansereau, 2017]."

**Lines 175-179:** "Note that a VP model is a nearly ideal plastic material, i.e. it can be considered as an elastic-plastic material with an infinite elastic wave speed; therefore, the fracture propagation is instantaneous (i.e., it is resolved with the outer loop solver of an implicit solver or the sub-cycling of an EVP model). In the above equation, n is a free parameter setting the steady-state damage for a given deformation state." See general comments above about distinguishing between the equations and the numerics and about the contradiction between including a damage evolution equation together with a steady-state stress equation, and actually including the concept of damage within a VP framework.

It is only a contradiction when we think about damage as a decrease in elastic modulus. In this sentence, we are simply stating that the damage is controlled by the viscosity.

**Line 316:** "soup-like": fluid-like?

Corrected as suggested by the reviewer.

**Line 317:** "except when th ≈ td when fewer extreme deformation events are present." See previous comment: This is rather trivial. At each time step, over completely-damaged elements (d = 1), you then offset damage by the same amount of healing: there is no net effect of damage nor healing.

This was clarified on line 317 of the revised manuscript: "In this case, the damage increment over partly damaged cells is compensated by the healing term."

**Lines 392-400:** Interesting. See general comments above for possible insights about time scales and temporal behavior.

We added this sentence after line 400: "The proposed damage parametrization simulates plastic weakening along LKFs and increases the memory of the system – something that the sea ice concentration and thickness do not accomplish when a lead opens and closes. It differs from the (elastic) damage definition introduced in earlier work (e.g., Girard et al., 2011; Rampal et al., 2016; Dansereau, 2016; Plante et al., 2020), where damage reduces the elasticity of the material, and therefore increases deformation for a fixed stress state. In the case of plastic damage, it is the effective viscosities in shear and divergence that are reduced, leading to larger strain rates for the same stress states. The VP sea ice rheology is one of many rheologies that simulates many climate-relevant sea ice properties satisfactorily (drift, deformation statistics) despite all of its flaws (yield curve, flow rule, elasticity simulated as viscous creep)."

---

## Author Response (AR2)

**Editor**

I agree the authors have gone some way to explaining the concept of plastic damage, which is what I would call plastic weakening. The exact language is immaterial as long as there is clarity. However, since damage is generally used in sea ice dynamics to refer to reduction of elasticity and not plastic weakening/damage, I think the title and abstract of this paper will still cause serious confusion. One of the referees strongly agrees with this. I will accept this paper if the abstract clearly defines what damage means in this paper, distinguishing it from elastic damage, and if the term "Plastic damaging" rather than merely "Damaging" is used in the title. If these changes are not made, then I will reject the paper and you should seek a different journal.

We changed the title and the abstract to accurately differentiate between the two concepts. The title now reads: *"On the sensitivity of sea ice deformation statistics to plastic damage"*. The first sentence of the abstract now reads: *"We implement a plastic damage parametrization (which is different from the elastic damage in the elasto-brittle framework) in the standard viscous-plastic sea ice model..."*

**Reviewer 1**

In their revised manuscript, the authors now argue that they introduced "plastic damage" within the VP modelling framework. This is a clarification, as this term (and associated references) were totally absent in the initial manuscript. Such "plastic damage", i.e., as a matter of fact a plastic weakening (as quoted by the editor in his review of the initial manuscript) or strain softening mechanism, has been indeed proposed previously, particularly in the concrete literature (e.g. Lubliner et al., 1989). However, I reiterate that this has very little to do with classical damage mechanics, introduced first by Kachanov. And, clearly, in the mechanical literature, damage, used as a single word, refers to a degradation of elastic stiffness, not to "plastic damage". Consequently, I think that the initial ambiguity is not totally removed in the revised manuscript, especially in the title. This is unfortunate, as this could have been done easily. In addition, a more complete conceptual setting could have been proposed from a comparison with these previous works, e.g. regarding the elastic-plastic strain decomposition assumption of Lubliner and associated papers, which is by construction irrelevant in the VP framework where elasticity is absent.

To further remove the ambiguity, we changed the title to: *"On the sensitivity of sea ice deformation statistics to plastic damage"*. Furthermore, we clarified the damage type that we use in the abstract: *"We implement a plastic damage parametrization (which is different from the elastic damage in the elasto-brittle framework) in the standard viscous-plastic sea ice model..."*

Finally, I consider that the answer of the authors to my comments about timescales (and, particularly between the "plastic damage" propagation timescale) is not really satisfactory, and I still believe that this will impact the representation of intermittency of sea ice mechanics. However, I would argue that it is an inherent flaw of the VP framework, with or without "plastic damage".

The damage timescale used in the model simulations is of the order of 1 day, in line with passive microwave satellite imagery that shows lead propagation over hundreds to a thousand kilometers over a one-day time scale. This time scale differs significantly from that used in the MEB model ( 1 sec, e.g. Dansereau, Weiss et al., 2016). First, Dansereau et al (2016) argue that the timestep must be *larger* than the damage advection time scale ($dx/c$, where $c$ is the elastic wave speed – see section 4.1.1). We disagree; the timestep must be much smaller (~2 orders of magnitude) than the advection time scale in order to resolve the damage propagation. Second, observations suggest that stress (and crack) propagation is slower than the elastic wave speed of the order of 1 km/s. We added a sentence on line 193 of the revised manuscript: *"The damage timescale used in the model simulations is in line with passive microwave satellite imagery that shows lead propagation over hundreds to a thousand kilometers over a one-day time scale..."*.

Regarding "intermittency". All sea ice models irrespective of the rheology show intermittency (Bouchat et al., 2023). So, clearly, the intermittency is not linked with elasticity, damage propagation and brittle fracture as argued in Weiss and Dansereau (2017). Answering this question correctly will require another study. In order to address this question, we intend to run a sea ice model in free drift, as a cavitating material (only resistance to compression), a material with resistance in compression and shear (with elastic stress included or not) and identify when intermittency appears in the sea ice deformation statistics.

**Reviewer 2**

Dear Antoine and Bruno,

While I disagree that using scaling information for validation of sea ice kinematics in models is a sensible tool, I feel your paper adds insight into how different models reproduce the statistical and scaling features that have been observed. For that reason, this is worth publishing. I still have reservations about encouraging the use of these metrics for evaluating models. Given the observations are not as clear cut and Marsan et al. (2004) suggests, I am not entirely convinced sea ice deformation has a well behaved multi-fractal nature. Perhaps the results in your models showing this, point towards quasi brittle systems not behaving as cleanly as Marsan et al. (2004) suggests. Also, what sets the scaling exponents is an interaction between forcing and dissipation that can be adjusted in multiple ways in models (and nature), which makes the use of this as a metric by itself for model tuning problematic. However in attempting to model the localization of deformation, this paper does add useful information to that conversation.

*The goal was to show that the deformation rate metrics are sensitive to the addition of a (plastic) damage parametrization. We agree that new metrics need to be developed to better evaluate the models.*

I would like to refer to reviewer 1 as to whether the discussion of the different damage parameterizations in EB and VP, and that they are not equivelent, is adequately addressed. Echoing the editor, we do not want to propagate misuse of standard terms.

**Specific comments**

A specific comment

In the introduction you state that models need to be both put into the same numerical framework to allow inter comparison. Do you actually achieve this? All your results are for the VP model, and not EB. So perhaps you need to reframe this to not mislead the reader you are doing this. But rather that you are focussed on the behavior of your modified VP model.

*We clarified this on line 106: "In an attempt to further disentangle the effect of elasticity, damage, and discretization, we include a plastic damage parametrization in the standard VP model, and analyze the deformation statistics of this modified VP model following recommendations from SIREx [ref], and [ref]. To this end, we compare both simulated (with and without damage in the VP model) and the RADARSAT-derived Eulerian deformation products using probability density functions (PDFs), spatiotemporal scaling laws, and multifractality."*

In reading the paper I found some consistent gramatical issues, that I suggest you proof read to make sure you catch throughout the manuscript. I point out below copy edit needs I found.

**Abstract line 4:** (as proposed in SIREx1) is not needed, and also SIREx should be expanded if this is kept.

*Corrected as suggested: we expanded the acronym.*

**line 8** "damage with a healing"

*Corrected as suggested by the reviewer.*

**line 9** "unveils"

*Corrected as suggested by the reviewer.*

**line 10** it might help to specify "sea ice deformation statitics" ... but your choice if you want to point the reader back to your not being general here. Sea ice is only mentioned in the first sentence of the abstract, and it is inferred that you continue to talk about this.

*Corrected as suggested by the reviewer. The sentence now reads: "...plastic damage parametrization is a powerful tuning knob affecting the deformation statistics of viscous-plastic sea ice."*

**line 11** The first sentence is hanging.

It is unclear to us as to why this sentence is hanging.

First paragraph of the introduction. One could argue that this whole paragraph is irrelevent to the topic of the paper. It is an interesting paragraph, and probably should be saved for the lead authors thesis. Though I would suggest adding some clarification to what you are describing.

We agree that this paragraph is not necessary for the paper in and of itself, but it eases the reader into the topic without being too long or impairing comprehension.

**Line 30** you use "sea-ice" and "sea ice" as an adjective in different parts of the text. Maintain a consistency with this, choose one convention.

Corrected as suggested by the reviewer throughout the paper.

**Line 43** not sure the use of hyphens here is good.

Corrected as suggested by the reviewer. We kept the hyphens but changed the phrasing such that it now reads: *"In the standard viscous-plastic (VP) rheology — with an elliptical yield curve and normal flow rule (e.g. ... and its variants) —, sea ice is..."*.

**Line 56** "Elasto-Brittle (EB) and modification thereof (MEB and BBM)"

Corrected as suggested by the reviewer.

**Line 58** "... larger deformations occur along fault lines ..."

This sentence already has a verb after fault lines.

**Line 62** check if deformation is singular or plural. This should be checked for correctness throughout.

Corrected as suggested by the reviewer throughout the manuscript.

**Line 365** and other places. You have a liking of starting a sentence with "Interestingly". Excessive use of this becomes uninteresting and distracts from your use of this construct to imbibe some interest in the reader.

Corrected as suggested by the reviewer throughout the manuscript. We replaced/removed multiple instances of "interestingly"

**Lines 442-444:** This sentence is mangled and hard to follow.

We added some precision about what specifically is changing: *"The VPd$(2, 3, 30, 27.5)$ configuration highlights a complex transient state in the multifractal behavior of the model when changing the parameter $n$ from fully damaged ice..."*

**Line 468-469** remove "– also called intermittency –"

Corrected as suggested by the reviewer

**Line 506** "... RGPS observations than that of ..."

Corrected as suggested by the reviewer

Code and data availability section. Citations for code and ice velocity model output not provided.

We published the code on Zenodo. The data is getting massaged to be published soon as well.